# Neural Bandit Based Optimal LLM Selection for a Pipeline of Tasks

## Abstract

With the increasing popularity of large language models (LLMs) for a variety of tasks, there has been a growing interest in strategies that can predict which out of a set of LLMs will yield a successful answer at low cost. This problem promises to become more and more relevant as providers like Microsoft allow users to easily create custom LLM "assistants" specialized to particular types of queries. However, some tasks (i.e., queries) may benefit from breaking down the task into smaller subtasks, each of which can then be executed by a LLM expected to perform well on that specific subtask. For example, in extracting a diagnosis from medical records, one can first select an LLM to summarize the record, select another to validate the summary, and then select another, possibly different, LLM to extract the diagnosis from the summarized record. Unlike existing LLM selection or routing algorithms, this setting requires selecting a sequence of LLMs, with the output of each LLM feeding into the next and potentially influencing its success. Thus, unlike single LLM selection, the quality of each subtask's output directly affects the inputs, and hence the cost and success rate, of downstream LLMs, creating complex performance dependencies that must be learned during selection. We propose a neural contextual bandit-based algorithm that trains neural networks that model LLM success on each subtask in an online manner, thus learning to guide the LLM selections for the different subtasks, even in the absence of historical LLM performance data. Experiments on telecommunications question answering and medical diagnosis prediction datasets illustrate the effectiveness of our proposed approach compared to other LLM selection algorithms.

## 1 Introduction

Large Language Models (LLMs) have transformed numerous applications with their ability to summarize, generate, and interpret text. Due to different underlying training configurations and model structures, LLMs can show a wide variation in terms of their performance for different tasks, raising the need to identify the best-performing model for a given task. However, with the sudden increase in the number of LLMs available and the complexity of tasks they are asked to perform, this challenge has become increasingly complex (Shnitzer et al., 2023). Indeed, some LLM operators even offer users the opportunity to build customized LLM agents, e.g., OpenAI's marketplace or Azure's Assistants (Microsoft, 2025), which may greatly expand the set of agents available to complete tasks. A natural question is then: **How should we select the best LLM agent to complete a given task?**

Selecting the most suitable LLM for a specific task or sequence of subtasks presents significant challenges, particularly in terms of computational efficiency and performance optimization, as has been explored in prior work (Zhang et al., 2024; Xia et al., 2024; Zhao et al., 2023). Due to computational, monetary, and latency concerns, simply running a query or task through all the available models and choosing the model that yields the best performance for that query is not feasible (Shazeer et al., 2017), while simply selecting the largest LLM for every task may be either prohibitively expensive (as these LLMs tend to be the most costly) or ignore the potential of highly specialized LLMs to perform well on specific types of tasks. These naïve approaches become particularly infeasible for **tasks that are too complicated or difficult for an LLM to handle alone**.

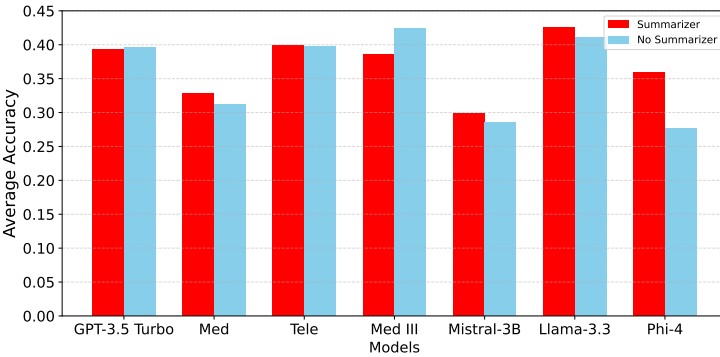

Figure 1: Average accuracies of LLMs (Med: GPT 4o finetuned on medical data, Tele: GPT 4o finetuned on telecommunications questions, Med III: GPT 4o finetuned on MIMIC III) on the medical diagnosis prediction task when the summarizer is chosen uniformly at random. Including a summarizer improves the average performance of most models, showing the value of breaking down a larger task into a pipeline of smaller ones. More detailed explanations about the models, experimental setup and results for this experiment are in Sec. 5.

These complicated tasks are generally broken down into multiple subtasks, each of which can then be completed by a different LLM, resulting in an exponential growth of possible LLM combinations, each of which will introduce a different success rate and monetary cost. Indeed, such use cases for LLM agents are only becoming more popular, with the rise of "agentic" AI for decomposing and completing complex, multi-step tasks (Qiu et al., 2024b). For example, Figure 1 shows the average accuracy of seven LLMs on the task of predicting a medical diagnosis from medical reports. We see that using a summarizer LLM first and then passing its output as the prompt to a diagnoser LLM, i.e., *using a pipeline approach, performs better than using a single LLM* for this task. Only the Med III model sees a visible drop in accuracy from using a summarizer in this task, which can be attributed to the fact that Med III is a finetuned GPT-4o model from the same dataset. *Using a summarizer also changes the optimal model choice for the diagnosis task*: while the Med III model performs best across all LLMs without summarizers, Llama performs best when a summarizer is used. Thus, the LLM selected for an earlier task in the pipeline may affect the optimal choice later in the pipeline, complicating decision-making. Similarly, LLMs selected for the summarization task can influence the *cost* of using different LLMs for the diagnosis task: the diagnoser cost will generally be proportional to the number of input tokens, which depends on the length of the summary provided by the chosen summarizer LLM. To the best of our knowledge, **ours is the first work to examine cost-effective LLM selection within such a pipeline of tasks**.

**Research Challenges.** In practice, users may not have historical data on LLM performance for specific types of tasks, especially if some candidate LLMs are customized "assistants." Thus, a natural solution is to take an *online approach*, where the user both learns and optimizes the LLM selection as it sends queries/tasks. Online LLM selection for a specific task can be viewed as an instance of the classical contextual multi-armed bandit (MAB) problem (Chu et al., 2011; Chen et al., 2013). Each LLM is modeled as an "arm," and the MAB algorithm sequentially pulls the arm (i.e., submits queries to the LLM), monitoring the query success and adapting future LLM selections accordingly. MAB balances exploration, i.e., trying new models in the hope that they outperform the identified current best model; with exploitation, i.e., selecting the best model found so far. Context, i.e., an embedding of the current query, is used to predict the success of each LLM on this query.

Conventional contextual MAB methods do not capture the *sequential* nature of LLM selection in our pipeline-of-tasks scenario: we must select one LLM for *each* subtask. Combinatorial variants of contextual MAB (Chen et al., 2018), in which a combination of arms (here, LLMs) are chosen in each round, is closer to our scenario, but it requires selecting LLMs for each task at once, instead of selecting the next LLM in the pipeline after observing the results from prior LLMs. Moreover, it is not clear how to modify these algorithms to account for both the success rate and the monetary cost of using each LLM. Thus, we propose a novel MAB variant designed specifically for *sequential* LLM selection, which we show outperforms conventional MAB methods.

The main **contributions** of this work are as follows:

- We introduce a novel **problem formulation** of selecting a pipeline of LLMs to solve a task decomposed into interconnected, smaller subtasks.

- We **adapt contextual MAB algorithms** into the Sequential Bandit algorithm, which sequentially selects LLMs in the task pipeline. Sequential Bandits utilizes neural networks to effectively learn each LLM's success at each subtask in the pipeline and optimize a combination of LLM success and cost in an online manner.

- To evaluate SeqBandits, we **create a new diagnosis prediction dataset** from an existing medical dataset (Johnson et al., 2016) of deidentified patients' medical reports.

- We **experimentally show** that SeqBandits identifies better LLMs than existing MAB algorithms for pipelined tasks on medical diagnosis and telecommunications datasets.

We first outline related work (Sec. 2) and describe our problem formulation (Sec. 3). We present our SeqBandits algorithm in Sec. 4 and its experimental evaluation in Sec. 5, then conclude in Sec. 6.

## 2  RELATED WORK

The proliferation of LLMs has led to growing interest in methods to predict the best-performing LLM. However, none of these methods fits completely into our setting. We divide these strategies into budget-aware frameworks, LLM cascades, and LLM routing.

**Budget Constrained Online Algorithms and Bandits**. These algorithms maximize cumulative reward subject to a hard cap on total resource consumption. Primal–dual schemes embed standard regret minimizers as black-box components to enforce long-term resource constraints via dual variables (Castiglioni et al., 2022), which can extend to integrate budgeted expert-query mechanisms that judiciously allocate a limited number of advice calls to sharpen decisions (Benomar et al., 2024). Non-stationarity and adaptive primal–dual updates have been shown to ensure constraint satisfaction even as cost and reward distributions shift over time (Liu et al., 2022). Most recently, weakly adaptive regret minimizers have been woven into primal–dual frameworks to simultaneously honor strict budget and return-on-investment limits (Castiglioni et al., 2024). Recently, contextual bandit approaches for multi-LLM selection under evolving contexts have been proposed, highlighting the difficulty of adapting model choices online when input distributions shift (Poon et al., 2025). Another work (Dong et al., 2024), considers single-step knowledge-based question answering with bandits, selecting a single model per question by using a cluster-level Thompson sampler and a linear contextual bandit with a cost-regret penalty. None of these, however, use sequential decision frameworks, as proposed in our setting.

**Cost-Efficient LLM Cascades**. In a typical cascading framework, inputs are processed through a pre-determined sequence of LLMs, from the least to the most resource-intensive. At each stage, the system evaluates the output to determine whether to accept the result or continue to the next model in the sequence (Zhang et al., 2024). Recent advancements have focused on integrating cascading with routing strategies, i.e., routing a query to the "best" LLM (Chen et al., 2023; Dekoninck et al., 2025b). Approaches like the Mixture of Thought representations combine chain-of-thought and program-of-thought prompts, in order to adaptively route simpler queries to smaller, less costly models, reserving more complex tasks for larger models (Cheng et al., 2023; Gao et al., 2023). This strategy has demonstrated significant reductions in inference costs while maintaining accuracy comparable to using the most robust LLM alone (Yue et al., 2024). However, cascading can be inefficient if the sequence is not optimally configured, as each input may need to pass through multiple models before reaching an adequate response (Dekoninck et al., 2025a). Unlike these cascading frameworks, Sequential Bandits considers a pipeline of LLMs, in which the different LLMs perform different subtasks that feed into each other (Li, 2025).

**Model Selection and Adaptive Routing**. Dynamic routing mechanisms, which intelligently direct queries to the most appropriate LLMs, can significantly improve performance and computational resources (Varangot-Reille et al., 2025; Somerstep et al., 2025), as unlike cascades, they do not make multiple passes through different LLMs. Systems like Tryage propose context-aware routing mechanisms that optimally select expert models based on individual input prompts (Hari & Thomson, 2023). This approach allows users to explore trade-offs between task accuracy and secondary goals like minimizing model size and improving response readability (Sikeridis et al., 2024). Other approaches like Zooter train a routing function using reward models' scores as supervision signals (Lu et al., 2023), efficiently directing queries to specialized LLMs (Chen et al., 2025).

Within the popular mixture-of-experts framework, Routing Experts introduces a dynamic expert scheme for multimodal LLMs that also aims to learn more efficient inference pathways (Wu et al., 2025; Saha et al., 2024; Liu et al., 2024). Other works focus on the challenge of dynamic routing, where new, previously unobserved LLMs become available at test time. These strategies generalize by representing each LLM as a feature vector (Jitkrittum et al., 2025). Frameworks like AutoMix (Aggarwal et al., 2025) focus on predicting LLM success, e.g., with a few-shot self-verification mechanism to estimate output reliability from smaller LLMs (Ding et al., 2024). Recent work like MixLLM and BEST-Route has expanded routing by introducing architectures that dynamically allocate queries across specialized LLMs, optimize test-time compute for accuracy–latency trade-offs, and adapt routing policies to real-time budget constraints (Ding et al., 2025; Wang et al., 2025). ADaPT (Prasad et al., 2024), introduces an adaptive task-routing framework where an LLM recursively decomposes a task only when the current executor is predicted to fail. Unlike standard routing, which selects a single model for the full query, ADaPT dynamically alternates between planning and execution to adjust task difficulty based on LLM capability. This yields more reliable performance on complex tasks. All of these methods, however, focus on selecting a single LLM per query, and cannot be directly applied to selecting a pipeline of LLMs as in our setting.

## 3 PROBLEM FORMULATION

We consider multiple rounds $t = 1, 2, \ldots$, where each round is defined by an incoming query $q_t$. In our formulation, we assume that the breakdown of a task into simpler subtasks $\{T_1, T_2, ..., T_k\}$ is given. These subtasks form a directed acyclic graph (DAG), as the output of a subtask becomes the input of the next task in the pipeline. For example, a query could be a medical diagnosis prediction based on a provided medical report, broken into the subtasks of (i) summarizing the report, (ii) validating the summary, and (ii) predicting a diagnosis given the validated summary. In the remainder of the paper, our aim is to select the LLM returning the highest accuracy for each subtask. Note that a special case of this setting is a query with one subtask (itself). Figure 2 illustrates our pipelined/sequential problem setting. This approach is in essence similar to the popular chain-of-thought (CoT) prompting (Wei et al., 2023), but we *allow different LLMs to complete each subtask*, unlike CoT in which a single LLM decomposes the task and completes each subtask. We formalize this selection problem below. Throughout, we denote by $[N]$, $N \in \mathbb{Z}^+$, the set $\{1, 2, \ldots, N\}$.

**Bandit Formulation.** In this paper, we model the LLM selection problem as a neural contextual bandit. We consider $T$ rounds. The subtasks $\{T_1, T_2, ..., T_k\}$ correspondingly have $\{N_1, N_2, ..., N_k\}$ available LLMs, each of which we refer to as an "arm" for a particular subtask. We use $\mathcal{M}$ to denote the set of all LLMs. In every round $t$, the agent selects an arm (LLM) for each of the subtasks corresponding to query $q_t$, and we name the overall set of arms selected as the **super arm** $S_t$, following the nomenclature of the combinatorial bandit literature (Chen et al., 2013). Formally, for each subtask $T_i$ where $i \in \{1, \ldots, k\}$,

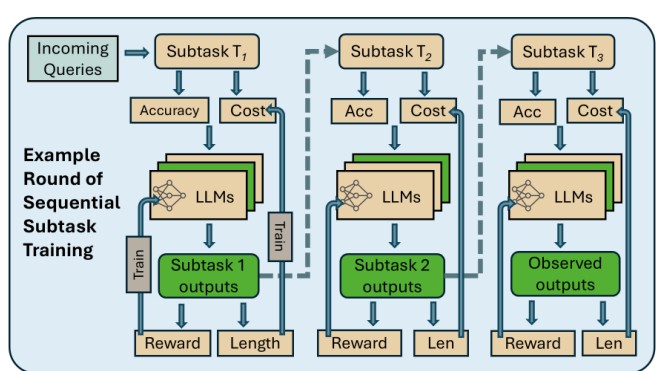

Figure 2: Our Sequential Bandits problem setting and approach, where an incoming query is split into simpler subtasks in a sequential pipeline. An LLM is chosen to complete each subtask and the output of an LLM is fed as the input to the next subtask's LLM in the pipeline.

select an arm $a_{i,j}$ where $j \in [N_i]$. Then $S_t = (a_{1,j}, a_{2,j}, \ldots, a_{k,j})$ where $a_{i,j}$ represents the arm $j$ selected from $[N_i]$ for subtask $i$. When a super arm $S_t$ is selected in round $t$, the agent observes the **base arm rewards** of the chosen super arm, namely $\{r_{t,j}\}_{a_{i,j} \in S_t}$ and receives a **total (super arm) reward** of $R(S_t, \mathbf{r}_t)$ where $\mathbf{r}_t = [r_{t,j}]_{a_{i,j} \in S_t}$, i.e., the super arm reward is a function of the individual base arm rewards. In our setting, the base arm rewards correspond to the "goodness" of the result for the subtask to which the arm belongs, and the super arm reward is a measure of

goodness of the LLMs' collective result on the query, which is a function of the selected base arm rewards. For example, the super-arm reward could be the accuracy of the output returned by the last LLM in the pipeline for a prediction task like medical diagnosis, which is the final output returned to the user. We also account for the **cost** associated with deploying the selected LLM for each subtask. While we take this to be the monetary cost, it could also be taken as the energy consumed or latency of the LLM's inference on the given subtask. We calculate the cost in terms of the number of input and output tokens of the chosen LLM for the given subtask and denote as $C_j(q_t)$ our predicted cost of using LLM $j$ for a subtask's query $q_t$. We give more detail about how $C_j$ is constructed in the next section.

**Context, Reward and Cost Estimation.** In every round $t$, the agent observes the **context** $\mathbf{x}_t(a_{i,j})$ of arm $a_{i,j}$ for each $i \in \{1, \ldots, k\}, j \in [N_i]$, which helps guide the prediction of LLM $j$'s success on subtask $i$. The context $\mathbf{x}_t(a_{i,j})$ in our application is a function of the given prompt for subtask $i \in \{1, \ldots, k\}$, which is the output of the LLM chosen for the previous subtask $i - 1$ as shown in Figure 2, as well as features of the LLM corresponding to arm $(i, j)$. In our experiments, these features include a description of the LLM's capabilities, as well as whether it was finetuned on a specific dataset for a certain task or if it is just a general-use LLM that is not finetuned. We will give more details on how we construct the context of the arms in the Supplementary Material. We denote the description of an LLM $j$ as $d_j$ and the overall set of descriptions of all available LLMs as $\mathcal{D}$.

Following prior works on neural bandits, we use a neural network(s) to learn the underlying reward function(s). We assume that $\forall t \in [T]$, the base arm reward $r_t(a_{i,j})$ of arm $a_{i,j}$ (i.e., LLM $j$ on subtask $i$) is generated as follows (Zhou et al., 2020; Zhang et al., 2021):

$$r_t(a_{i,j}) = h_{i,j}(\mathbf{x}_t(a_{i,j})) + \xi_t \tag{1}$$

where $h_{i,j}$ is the underlying unknown reward function for arm $a_{i,j}$ and $\xi_t$ is zero-mean noise, modeling uncertainty in the "goodness" of the LLM output. We allow separate reward functions for the different (subtask, arm) combinations, since each combination corresponds to a different model structure and task. To learn the base arm reward function $h_{i,j}$, we use a fully connected neural network with depth $L + 1 \geq 3$ and width $n$ (Zhou et al., 2020; Zhang et al., 2021):

$$f_{i,j}(\mathbf{x}; \boldsymbol{\theta}) = \sqrt{m} \mathbf{W}_{i,j}^{(L)} \sigma(\mathbf{W}_{i,j}^{(L-1)} \sigma(...\sigma(\mathbf{W}_{i,j}^{(0)} \mathbf{x})) \tag{2}$$

where $\sigma(x) = \max\{x, 0\}$ is the ReLU activation function and $\boldsymbol{\theta}_{i,j} = [\text{vec}(\mathbf{W}_{i,j}^{(0)})^T, ..., \text{vec}(\mathbf{W}_{i,j}^{(L)})^T]^T$ is the weight vector. We denote the weight vector in round $t$ for arm $a_{i,j}$ as $\boldsymbol{\theta}_{i,j}^t$. We let the gradient of the neural network be $\mathbf{g}_{i,j}(\mathbf{x}; \boldsymbol{\theta}_{i,j}) = \nabla_{\boldsymbol{\theta}_{i,j}} f_{i,j}(\mathbf{x}; \boldsymbol{\theta}_{i,j})$.

The cost $C_j(q_t)$ for LLM $j$ is modelled as the number of input and output tokens, multiplied by LLM $j$'s corresponding per input token and output token costs in Microsoft Azure (see the Supplementary Material for more details). We know the number of input tokens since $q_t$ is given. Since we do not know the number of output tokens LLM $j$ will output in advance for a given prompt, we train an output token length prediction model as in Qiu et al. (2024a), as detailed in the Experiments section. We denote the output token length prediction model's output for LLM $j$ at round $t$ with $c_j(t)$.

**Agent Objective.** The main objective of the agent in this setting is to maximize the reward (accuracy) while also minimizing the cost of deploying LLMs for the tasks. To combine these two metrics into a single one, we define the **net reward** at time $t$ as $N(t) = R(S_t, \mathbf{r}_t) - \boldsymbol{\alpha} \cdot \mathbf{C}(S_t)$ where $\boldsymbol{\alpha} = [\alpha_1, \alpha_2, ..., \alpha_k]$ trades off the relative importance of reward and cost for each subtask and $\mathbf{C}(S_t)$ is the vector of costs associated with the chosen LLMs. Maximization of reward also leads to the minimization of **regret**, a standard MAB metric that measures the gap in reward between the optimal and selected arms. More formally, the regret $\mathcal{R}(T) = \sum_{t=1}^{T}(R(S_t^*, \mathbf{r}_t^*) - R(S_t, \mathbf{r}_t))$, where $S_t^*$ represents the optimal super arm at round $t$, which is the combination of arms that yield the highest reward, and $R(S_t, \mathbf{r}_t)$ is the reward of our chosen super arm at round $t$.

## 4 SEQUENTIAL BANDITS ALGORITHM

In this section, we present our proposed algorithm, Sequential Bandits (Algorithm 1), which aims to maximize the net reward. Sequential Bandits is a neural network based contextual bandit algorithm that initializes a neural network for every (subtask, LLM) combination. Figure 2 illustrates how these networks are trained using reward feedback, which is formally described in Algorithm 1.

The task is first divided into simpler subtasks. For the first subtask, for every available LLM, we construct an upper confidence bound (UCB) on the reward using the neural network estimate for exploitation and its gradient for exploration (line 6), similar to Zhou et al. (2020). Here we denote the $\ell_2$ norm by a positive definite matrix $\mathbf{A}$ by $\|\mathbf{x}\|_{\mathbf{A}} := \sqrt{\mathbf{x}^T \mathbf{A} \mathbf{x}}$. However, unlike Zhou et al. (2020), our algorithm includes a cost sensitivity parameter $\alpha_i$, which is multiplied by the cost term $C_j$, the predicted cost of using the chosen LLM $j$ for subtask $i$, and then subtracted from the UCB term. Setting $\alpha_i = 0$ reduces to the cost-agnostic setting. Since the cost term is subtracted from the UCB term in line 6, this leads to a tradeoff between the accuracy and cost. As the value of $\alpha_i$ increases, the algorithm will prefer cheaper models, prioritizing cost over accuracy for task $i$.

As described in Sec. 3, the input to the neural network includes embeddings of the description $d_j$ of each LLM $j$, as well as the incoming query $q_t$. We then choose the LLM that has the highest reward estimate (line 7) and pass the output of the chosen LLM as the input prompt for the next subtask in the pipeline (lines 8-9). For the other subtasks, we follow the same steps, except that instead of the input query $q_t$, the input prompt for the LLMs in the next stage of the pipeline (next subtask) is the output from the previously chosen LLM (lines 11-14). After choosing an LLM for every subtask, we observe their corresponding rewards and the overall super arm reward $R(S_t, \mathbf{r}_t)$, a function of the subtask rewards (line 17) and the token length of the output, which is used to train the token length prediction model $c(t)$ (line 10, 16). We then update the weights of the chosen LLMs' corresponding networks using the observed rewards, as well as the exploration parameter (lines 18-19).

**Incorporating Costs.** An alternative method of incorporating the cost term would impose a cost constraint (e.g., a monetary budget per query or task) instead of incorporating it into the objective. However, *LLM inference costs are inherently uncertain before query execution*, due to their dependence on the number of output tokens. Thus, a hard budget constraint could favor subtasks early in the pipeline, as underestimations of their costs could then lead to severe budget constraints for later subtasks. Incorporating cost into our objective also allows us to tune $\alpha_i$ for different subtasks $i$, which may have different inherent costs: for example a summarization task may be quite expensive as its input consists of a long text to be summarized.

**Differences from Existing Neural MAB Algorithms.** Prior neural contextual bandit algorithms (Zhou et al., 2020; Xu et al., 2020) can be naïvely adapted to our problem setting by simply using them for each subtask's LLM selection in sequence, which we use as baselines in Sec. 5. However, such algorithms train and use *one* neural network for reward estimation and LLM selection for each subtask. Using this single model for all LLMs does not account for the fact that different models imply a different inherent reward function for the different subtasks for each LLM. Thus, Sequential Bandits uses a *separate* neural network for every (subtask, arm) combination. Experimentally, we find that *using a separate neural network encourages more exploration*, as using the same neural network estimator for each (subtask, arm) combination leads to more similar success estimates. The relative ranking of the different arms is then dominated by the (deterministic) cost estimates, limiting exploration and increasing regret. We note that *using multiple neural networks does not increase our training overhead*: while we use different networks for every (subtask, LLM) combination, in each round we only train the neural networks of the LLMs that we have chosen for every subtask. Thus, our required compute is no greater than the other neural contextual MAB algorithms.

## 5 EXPERIMENTS

In this section, we present the results of our proposed Sequential Bandits algorithm on two use cases: medical diagnosis prediction and telecommunications question answering tasks. We compare our algorithm with the following baselines: (1) **Random**, which randomly selects an LLM for each subtask; (2) **Llama** or **Tele**, which always selects Llama or Tele for the given subtasks (Llama is chosen as it is the best performing model across tasks for the medical setting and Tele is the best for the telecom setting); (3) **Cost-Aware NeuralUCB** (Zhou et al., 2020), which is the cost sensitive version of NeuralUCB that uses a neural network for each subtask's reward prediction and adds the same weighted cost term in the objective as used in Sequential Bandits; and (4) **Cost-Aware NeuralLinUCB** (Xu et al., 2020), the cost-sensitive version of NeuralLinUCB, which makes use of neural networks for each subtask to learn representations of the contexts, and applies a linear model on these learned features to predict the rewards, then adds the weighted cost term to the objective; (5) **Cost-Aware NeuralUCB Joint**, which makes use of neural networks for each subtask as (3) but se-

---

**Algorithm 1** Sequential Bandits

---

1: Input: LLMs $m \in \mathcal{M}$, descriptions $d \in D$, queries $[q_1, q_2, \ldots, q_T]$, number of gradient descent steps $J$, learning rate $\eta$, cost weight parameters $\alpha_i$, number of rounds $T$, regularization parameter $\lambda$, initialize $\mathbf{Z}_0(a_{i,j}) = \lambda\mathbf{I}$ for all arms $a_{i,j}$.
2: **for** $t = 1, \ldots, T$ **do**
3:     Observe descriptions $d \in D$ and subtasks $i \in \{1, 2, \ldots, |\mathcal{T}|\}$
4:     **for** subtask $i = 1, \ldots, |\mathcal{T}|$ **do**
5:         **if** $i = 1$: **then**
6:             Compute $\forall j \in [N_i]$, $u_{i,j} = f_{i,j}(q_t, d_j) + ||\mathbf{g}_{i,j}(\mathbf{x}_t(a_{i,j}); \boldsymbol{\theta}_{i,j}^{t-1})/\sqrt{n}||_{\mathbf{Z}_{t-1}^{-1}(a_{i,j})} - \alpha_i C_j(q_t)$
7:             $s_i = \mathrm{argmax}_j(\mathbf{u}_i)$ (LLM chosen for subtask $i$)
8:             Pass query $q_t$ through LLM $s_i$
9:             Observe output $p_{i+1}$ of the chosen LLM
10:           Train token length prediction model $c(t)$ using observed output $p_{i+1}$
11:         **else**
12:             Compute $\forall j \in [N_i]$, $u_{i,j} = f_{i,j}(p_i, d_j) + ||\mathbf{g}_{i,j}(\mathbf{x}_t(a_{i,j}); \boldsymbol{\theta}_{i,j}^{t-1})/\sqrt{n}||_{\mathbf{Z}_{t-1}^{-1}(a_{i,j})} - \alpha_i C_j(p_i)$
13:             $s_i = \mathrm{argmax}_j(\mathbf{u}_i)$ (LLM chosen for subtask $i$)
14:             Pass prompt $p_i$ through LLM $s_i$
15:             Observe output $p_{i+1}$ of the chosen LLM
16:           Train token length prediction model $c(t)$ using observed output $p_{i+1}$
17:         **end if**
18:     **end for**
19:     Play super arm $S_t$ and observe rewards $\{r_{t,j}\}_{a_{i,j} \in S_t}$ and super arm reward $R(S_t, \mathbf{r}_t)$
20:     For selected arms $a_{i,j} \in S_t$, update $\mathbf{Z}_t(a_{i,j}) = \mathbf{Z}_{t-1}(a_{i,j}) + \sum_{a_{i,j} \in S_t} \mathbf{g}_{i,j}(\mathbf{x}_t(a_{i,j}); \boldsymbol{\theta}_{i,j}^{t-1})\mathbf{g}_{i,j}(\mathbf{x}_t(a_{i,j}); \boldsymbol{\theta}_{i,j}^{t-1})^T/n$
21:     Update weights of selected arms $\boldsymbol{\theta}_{i,j}^t$ by minimizing the MSE loss using gradient descent with step size $\eta$ for $J$ iterations by using the reward feedback.
22: **end for**

---

lects LLMs for subtasks all at once rather than sequentially, and (6) **FrugalGPT** (Chen et al., 2023), which is a offline cascade-construction algorithm that we integrated with our own prompts, API calls, online evaluation pipeline (hence how we generate cumulative reward curves), and datasets to ensure fair comparison. We adapted FrugalGPT's cascade-construction module by replacing its internal response completion and scoring with our own Azure-based inference wrapper and dataset specific reward function. We also implemented a round-based evaluator that repeatedly applies the static cascade to the query sequences, allowing us to compute the cumulative reward, cost, and regret curves even though FrugalGPT itself performs no online learning. Additionally, since difference cascade strategies are trained based on the allowed budget, for fairness we constrained the budget to be 3 times that our of diagnosis subtask cost. Comparisons to these algorithms respectively demonstrate the value of (1) having a non-static algorithm that learns over time, (2) intelligently selecting a LLM for each subtask, (3,4) using a separate neural network to predict each LLM's performance on each subtask, (5) sequentially selecting LLMs, and (6) having an online learning algorithm.

## 5.1 Experiment Settings

We evaluate Sequential Bandits on two **datasets**: one created from MIMIC-III (see the next subsection), a comprehensive clinical database containing de-identified health-related data from over 40,000 critical care patients (Johnson et al., 2016); and TeleQnA, a dataset comprising 10,000 multiple-choice questions designed to assess LLMs' knowledge in telecommunications (Maatouk et al., 2023). We thus assess our framework's performance across diverse, domain-specific tasks. We use pipelines with 2 and 3 subtasks respectively for the medical and telecommunications datasets.

**Subtasks.** Our two subtasks on the medical dataset are (1) a summarizer that summarizes the long medical report, whose summary is then fed to (2) a diagnoser that gives a diagnosis based on the summary. The **rewards** for the summarization subtask are obtained using an evaluation LLM that is fed the prompt, context, and benchmark. The reward for the diagnosis subtask given the summarized

report compares the diagnoses that the LLM outputs to the actual diagnoses of the patient, which also serves as the super arm reward. For example, if a patient has two diagnoses, and the second subtask's LLM correctly predicts one of the two diagnoses of the patient, we assign a reward of 0.5.

For TeleQnA, we have a 3 subtask structure. We start with a summary subtask with a similar reward metric as the medical dataset's summary task. The second subtask is to answer the question, for which we compute the LLM rewards by comparing their output choice (among the 4-5 options) to the correct one. The third subtask was to explain the answer obtained from the previous subtask, whose reward is obtained by comparison to TeleQnA's explanation benchmark.

**Cost Prediction.** For the output token length prediction model, which is used to construct the expected cost ($C_j$ in Algorithm 1) of LLM query execution, we first train an offline Bert regression model on the LMSYS-Chat-1M dataset (Zheng et al., 2023) using L1 loss, following Qiu et al. (2024a). We use 50,000 examples with a 90% to 10% training and validation split, and provide the training and validation loss curves for the offline training in the Supplemental. This model is also updated throughout the online training loop and takes an optimizer step after observing several input token length-output token length pairs for the LLM models chosen for the subtasks. We include online training results for this model in Figures 12, 13. We provide more information about the offline training of this model and the loss curves in the supplementary material.

All of the **models** we used in our experiments were deployed on Microsoft Azure. The models include a combination of base models (GPT-3.5-turbo, GPT-4.1 mini, GPT-4o, Llama-3.3-70B-instruct, Mistral-3B, Phi-4), finetuned models that were finetuned using Azure on general medical and telecom knowledge, and GPT-3.5-turbo assistants that used file search prompted with relevant field knowledge. The models were selected in a way such that there are low performing/SLMs, base models, and finetuned/custom domain knowledge possessing LLMs (Figure 1). However, as the LMSYS-Chat-1M dataset (Zheng et al., 2023), which we use to train our token length prediction model, does not contain the Mistral and Phi models, we omit them in the experiments presented in here but include them in the Supplementary's cost-agnostic ($\alpha = 0$) results. The models ordered from cheapest to most expensive are: GPT-4.1 mini, Llama 3.3, Med, Tele, Med III.

The n2c2 smoking dataset (Uzuner et al., 2007) was used to fine-tune the Med model, with individual reports constructed into input/label pairs by extracting the diagnoses embedded within the reports. Given the format of the medical reports, they were initially processed through a summarizer to extract general medical knowledge before being utilized to fine-tune the GPT-4o models. The other two fine-tuned models, Med III and Tele, were respectively fine-tuned with the MIMIC-III and TeleQnA datasets. **Evaluations** were performed using the Microsoft Azure evaluator, and the fine-tuned models generally performed better than the base models on the task they were tuned on.

## 5.2 DIAGNOSIS PREDICTION DATASET

There are some widely available medical datasets that include medical reports for de-identified patients. However, to our knowledge, there is no available dataset that is specifically tailored towards diagnosis prediction based on medical reports. Hence, we developed our own dataset from MIMIC III (Johnson et al., 2016) to be able to successfully assess Sequential Bandits' performance.

Johnson et al. (2016)'s MIMIC III dataset contains the admission diagnoses as well as diagnoses identified later, along with details of patient stay in their reports. Each patient has a number of reports based on their duration in the hospital, which we combine into a single report for each patient. We remove all explicit mentions of the patient's identified diagnoses, as well as diagnosis-related comments. We include the observations made and the test results of the patients in the reports. Patients we included in the dataset were mainly diagnosed with diseases related to the heart, kidneys, liver, and brain. Our dataset includes 100 medical reports with corresponding diagnoses. We include a more comprehensive list of the diagnoses of the patients in the Supplementary Material and some example patient reports. The full dataset is available in our released code.

## 5.3 EXPERIMENTAL RESULTS

We now present the experimental results for our medical diagnosis dataset and the TeleQnA dataset. All algorithms were run ten times and five times respectively for the medical and telecommunication settings, with tuned hyperparameters. The shaded regions in Figure 3 indicate standard deviations.

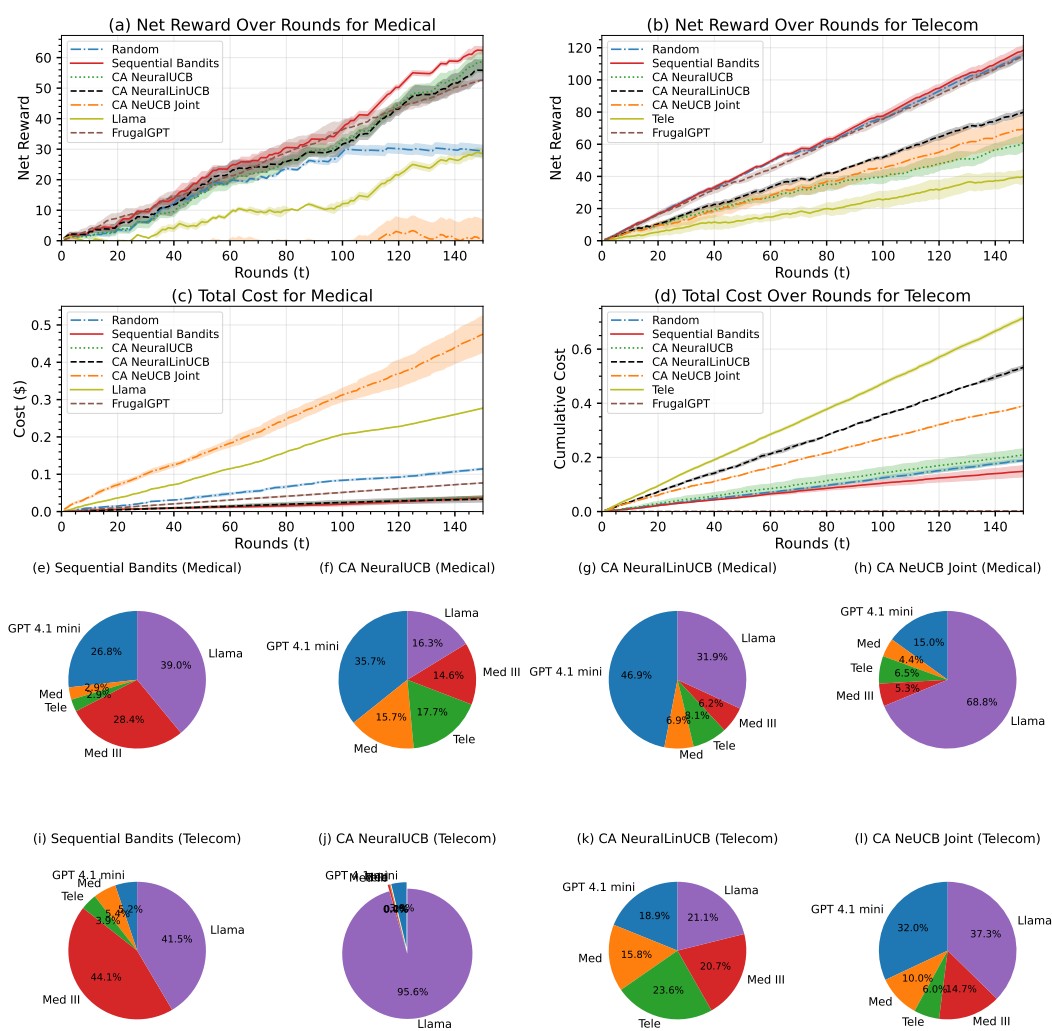

Figure 3: (a) Net reward of our algorithm compared with other baselines for the medical pipeline (b) Net reward of our algorithm compared with other baselines for the telecom pipeline (c) Total cost incurred by the algorithms for the LLM selections they make for the medical pipeline (d) Total cost incurred by the algorithms for the LLM selections they make for the telecom pipeline (e)-(h) Model selections for the diagnosis subtask in medical experiments (i)-(l) Model selections for the explanation subtask in telecom experiments

**Net Reward and Cost**. We first show that *Sequential Bandits obtains a higher net reward $N(t)$ than baseline algorithms on both pipelined settings*. For the medical setting, we consider the reward to be the reward of the final subtask in the pipeline (i.e., diagnosis accuracy) while for the telecom setting, the reward is the sum of the reward for the explanation and multiple choice answer. Figure 3 shows that our algorithm (solid red lines) achieves the highest net reward and also among the lowest costs, in both settings. When comparing the final net reward, Sequential Bandits has a %7.20 improvement over the most competitive baseline (CA NeuralUCB) in the medical setting while displaying a %7.27 improvement over the most competitive baseline (Random) in the telecom setting. In fact, when we look at Figure 1, which displays the diagnosis accuracy for the different models in the medical setting, we would expect Llama to outperform Sequential Bandits, as Llama has the highest accuracy among the available models in this setting. However, the fact that our algorithm outperforms Llama shows that it can learn the complex dependencies between the subtasks, as there may be combinations other than Llama for both summarizer and diagnosis that obtain a higher net

reward. The FrugalGPT baseline obtains the lowest cost among all baselines for the telecom setting, mainly because it doesn't use a pipeline, but it still gets a lower net reward than Sequential Bandits. For the telecom dataset, the Tele baseline performs the *worst* among all the baselines in terms of net reward, despite the fact that it is fine-tuned on this dataset, since it incurs a much higher cost ($\approx \$0.6$). Random performs the worst in the medical setting while it surprisingly performs well in the telecom setting.

**Analyzing LLM Selections.** We finally take a closer look at which LLMs are selected by each algorithm for the medical diagnosis subtask and telecom explanation subtask respectively. Figure 3(e) shows that Sequential Bandits selects Llama, Med III and GPT 4.1 mini the most often (%39, %28.4, %26.8 respectively), which enables it to have high net reward and low cost as these are cheap models, and Llama and GPT 4.1 mini also have the highest and second highest accuracy for this subtask respectively. Even though Med III is also selected often, it gives less verbose outputs and just its predicted diagnosis, contributing to low cost. The other baselines, as shown in Figure 3(f)-(h), select these models less frequently and make more suboptimal choices such as selecting the Med model more often, which has the lowest accuracy. CA NeUCB Joint selects similar models to our algorithm for diagnosis, but it selects different summarizers from Sequential Bandits which results in the performance difference. For the telecom explanation subtask seen in Figure 3 (i)-(l), Sequential Bandits selects Med III and Llama the most, which achieve the highest and second highest accuracy for this task respectively (Figure 21) and selects the remaining models relatively equally. CA NeuralUCB selects Llama the most while CA NeuralLinUCB selects all models nearly equally. CA NeUCB Joint on the other hand, shows a preference towards Llama and GPT 4.1 mini. Additional pie charts for the telecommunications and medical settings, indicating the model choices, can be found in the Supplementary Material.

**Scalability and Runtime.** A possible concern that might arise is whether learning separate neural networks for each subtask (or (subtask, LLM) pair in Sequential Bandits) would increase the runtime and required compute compared to other baselines. We want to note that as Sequential Bandits is only training the corresponding network of the chosen model for each subtask, it's doing the same amount of training as the other neural bandit baselines. We have also compared the runtime of our algorithm with the Random baseline, which does not train any networks during training. Random achieved an average runtime of 5657 seconds per run on the Telecom dataset over 150 rounds whereas Sequential Bandits finished in 5666 seconds per run over 150 rounds averaged over 5 runs. This result shows that the main bottleneck of the runtime are the API calls that we make rather than the online training as both achieve similar runtime. The neural networks that we train are also relatively lightweight, with a total parameter count of $\approx 19300$ for each of the networks.

**Additional Experiments.** We include more experimental results in the Supplementary Material, including (i) the regret achieved by each algorithm and model accuracies on subtasks. We also present results for (ii) a 3 subtask version of the 2 subtask medical pipeline in the main paper, and (iii) cost-agnostic ($\alpha = 0$) versions of the Medical Diagnosis task, as well as a single-subtask TeleQnA task. Since the cost can also be interpreted as response latency, we further present results on the model response latencies, showing they are hard to predict.

## 6 CONCLUSION

In this paper, we introduced a novel approach to selecting a pipeline of LLMs for executing decomposed tasks, employing a contextual MAB algorithm that sequentially chooses the best LLM for each sub-task in an online manner. Our approach, Sequential Bandits, leverages neural networks to model the expected success of each LLM, thereby enabling more effective task completion across a sequence of dependent subtasks. We demonstrated the effectiveness of our method through experiments involving medical diagnosis prediction and multiple choice telecommunications question answering. Our work provides a framework that can be adapted to various domains where task decomposition is feasible. Future work includes handling a broader range of tasks with varying degrees of difficulty and interdependencies, and providing regret guarantees or other performance bounds, as well as incorporating the task decomposition itself into the online learning framework using an approach similar to ADaPT (Prasad et al., 2024).

**Reproducibility Statement** We have taken several steps to ensure the reproducibility of our work. The full problem formulation, algorithmic details, and pseudocode for Sequential Bandits are pro-

vided in the main text (Secs. 3–4) and Algorithm 1. Details about hyperparameter tuning and other experiment settings, e.g., formation of the contexts using embeddings, may be found in the Supplementary Material. All datasets we used are either public datasets mentioned in the experiment settings or available in our code repository. We also release our training and evaluation scripts to reproduce the main results and figures. Evaluation metrics, baseline implementations, and cost estimation methods are fully described in Sec. 5 and the Supplementary Material. Together, these materials provide the necessary resources to reproduce our findings.

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

# A    Cost Agnostic Setting Results

First, we present our experimental results for the cost-agnostic versions of our algorithm for a non-pipelined single task telecommunications setting for the TeleQnA dataset Maatouk et al. (2023) and for the cost-agnostic version of the 2 subtask medical pipeline we presented in the main paper. Here all the algorithms considered (NeuralUCB and NeuralLinUCB) are also cost-agnostic versions. As we are in the cost agnostic setting, $\alpha = 0$ hence net reward reduces to reward in this setting.

## A.1    Results for the Telecommunications setting

In this section, we present results on the non-pipelined multiple choice telecommunications question answering task. In this setting, a single LLM is chosen which is given as input the telecommunications question and outputs one of the available choices. Now, we show the reward and regret plots for this setting. Figures 4 and 5 show the cumulative regret and reward and shows that our algorithm outperforms the baselines. The error bars indicate the standard deviation which was obtained by averaging over 5 runs.

Next, we present the average accuracies of the LLMs for this task. The averages were taken over 15 runs. The accuracies in this setting is binary at every round, since in each round there is an incoming multiple choice telecommunications question which has only one correct choice. Figure 6 shows the accuracies for the different LLM models, indicating that the fine-tuned Telecom model from GPT-4o is the best performing one, followed by GPT-3.5 Turbo, Med III, and Med models. As in the diagnosis prediction task, Mistral-3B achieves the lowest accuracy for this task.

The pie charts that can be seen in Figure 7 show the percentage of model selections for the telecommunications task averaged over 5 runs. Random exhibits a relatively even selection of the models as expected while the remaining algorithms select the Telecom model the most. Our algorithm, Sequential Bandits plays the Telecom model the majority of the time (52.2%) which is the highest among all algorithms. Compared to the application of diagnoses prediction from medical reports, NeuralUCB and NeuralLinUCB perform better, as indicated by their model selections as they are able to identify the models with higher accuracy and play them more.

## A.2    Results for the Medical setting

Now, we will present the results for the 2 subtask medical cost-agnostic setting. This is the same medical setting as we presented in the main paper where we have a summarizer LLM that summarizes the input medical report which is then passed to a diagnoser LLM that makes a diagnosis based on this summarized report. The main difference is that now we are in the cost-agnostic setting, meaning that we only look at the predicted accuracy of the models and not their cost. We consider two different types of random selection algorithms in these experiments. Random, selects an LLM at random to complete the diagnosis prediction task given the unsummarized medical report while SeqRandom operates under random selection for the 2 subtasks (equivalent to Random in the main paper).

As can be seen from Figures 8 and 9, Sequential Bandits achieves the highest reward while also achieving the lowest regret among the baselines. The pie charts in Figure 7 show the percentage of models each algorithm selects for the diagnosis prediction subtask. As expected, Sequential Random shows a roughly uniform selection among the models. NeuralUCB, surprisingly, seems to exhibit a similar trend despite using a neural network to predict LLM success; however, this is because these pie charts show the average model selections among five individual experiments. In each experiment, NeuralUCB fixated on a different, usually suboptimal, LLM, causing the average to appear near-uniform across the LLMs. Our algorithm, Sequential Bandits, shows a preference toward the Med III and Telecom models, while selecting Llama-3.3, GPT-3.5 Turbo and Phi-4 relatively equally. Notably, our algorithm selects Mistral-3B and Medical very sparingly (both 2.1%), which are the two worst performing models for this subtask, as indicated by the red bars in Figure 1. NeuralLinUCB also selects Mistral-3B rarely (4.9%); however, it selects the Medical model quite often (20.6%).

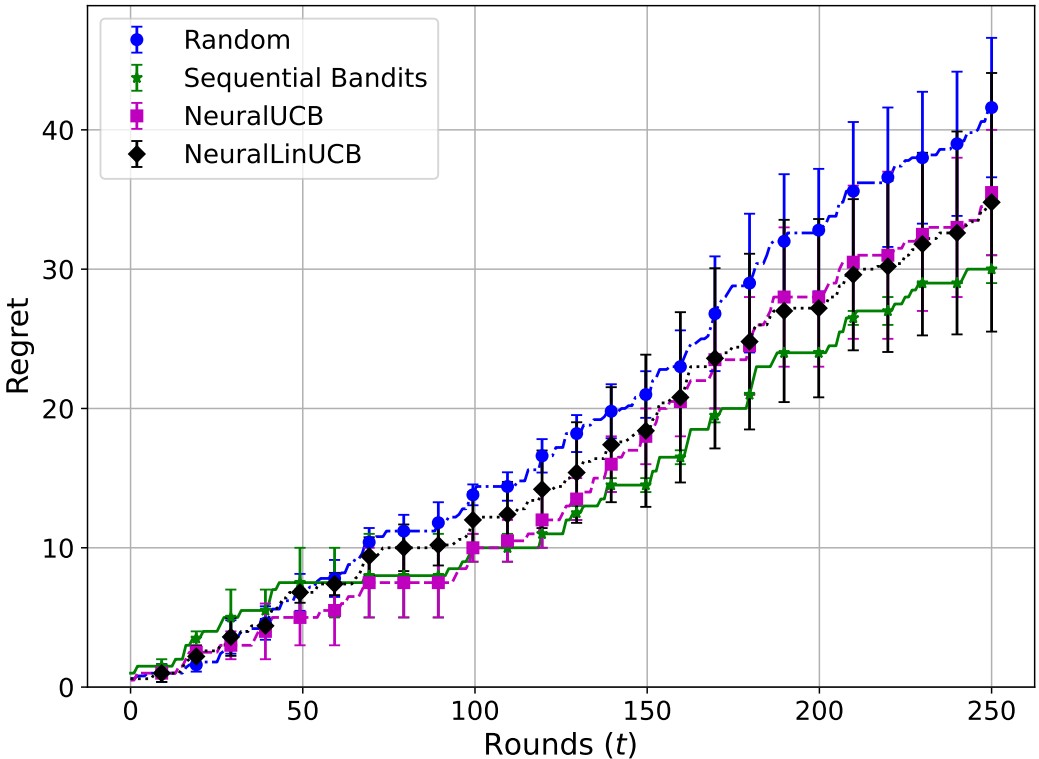

Figure 4: Cumulative regret for TeleQnA in the single task cost agnostic setting

## B    EXPERIMENTAL SETUP, HYPERPARAMETER TUNING AND DIAGNOSIS PREDICTION DATASET

### B.1    EXPERIMENTAL SETUP AND HYPERPARAMETER TUNING

We now detail the hyperparameters we tuned for all the algorithms and how we formed the contexts for the different tasks from the input prompts and the descriptions of the LLMs. To get embeddings for the prompts and the descriptions we use HuggingFace's "Paraphrase-MiniLM-L6-v2" Sentence Transformer model which embeds into 384 dimensional vectors. We take the elementwise multiplication between the description of an LLM and its input prompt to form the context for the (LLM, subtask) combination. Elementwise multiplication is a commonly used similarity metric which is why we also utilize it in our experiments. This context is fed as input to the neural networks, which are 2-layer fully connected networks with a width of $n = 50$. We set the regularization parameter $\lambda = 1$ for all experiments and tune the learning rate $\eta$ between $10^{-2}$ and $10^{-4}$. We set the number of gradient descent steps as $J = 5$. We also tune the ratio between the exploitation term (mu) and the exploration term (sigma) from 0.1-2 for the algorithms to balance exploration and exploitation as optimally as possible. Finally, we also tune the cost-sensitivity parameter $\alpha$ for all our experiments. We tuned this parameter to ensure that the magnitude of the cost term and the accuracy term in the objective in Algorithm 1 is roughly equal to each other so that we give similar importance to cost and accuracy. The $\alpha$ values range from 50-150 in our experiments.

### B.2    DETAILS ABOUT OUR DIAGNOSIS PREDICTION DATASET

We now give more details about the diagnostic prediction dataset that we created, listing the diagnoses of the 100 patients, and giving an example medical report. The medical reports for all 100 patients along with their corresponding diagnoses can be found in the supplementary material file. The diagnoses of the patients include the following: congestive heart failure, coronary artery disease,

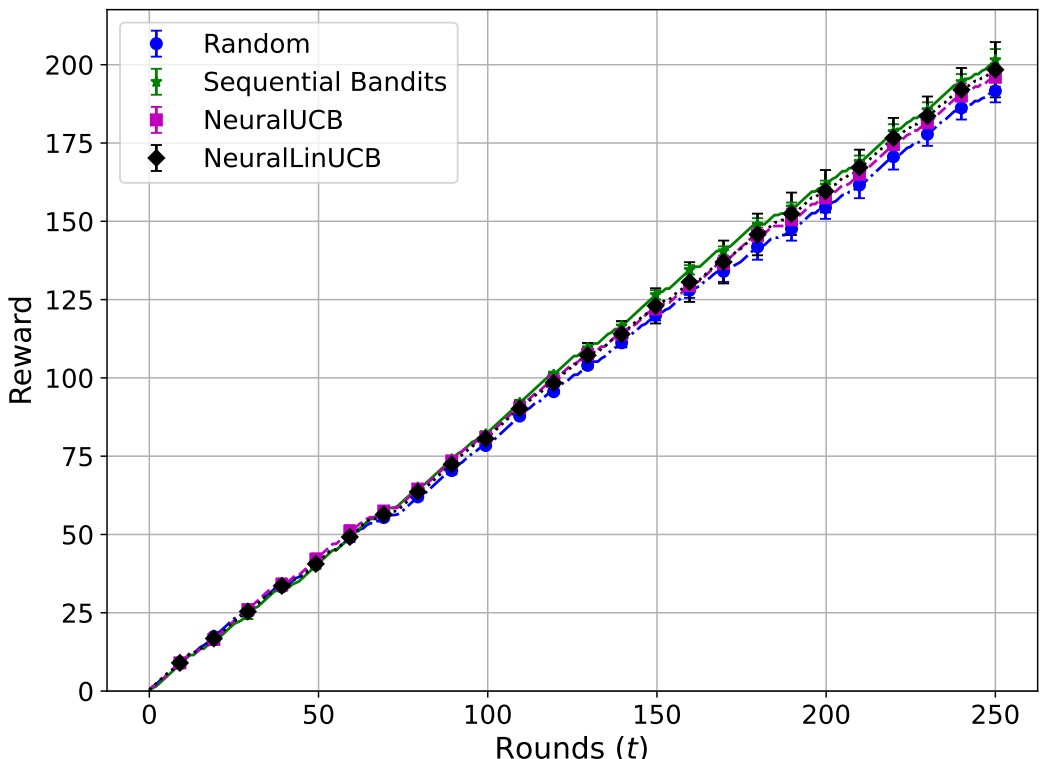

Figure 5: Cumulative reward for TeleQnA in the single task cost agnostic setting

sepsis, acute coronary syndrome, v-tach, unstable angina, acute myocardial infarction, aortic stenosis, pulmonary embolism, pneumonia, opiate intoxication, carotid stenosis, complete heart block, respiratory failure, liver failure, encephalopathy, tamponade, neurosarcoidosis, aortic dissection, abdominal thoracic aneurysm, svc syndrome, urosepsis, hyponatremia, hypoxia, aortic insufficiency, cirrhosis of liver, vertebral/basilar stenosis, renal failure, chronic obstructive pulmonary disease, urinary tract infection, pulmonary edema. An example medical report for a patient is as follows:

FINAL REPORT TECHNIQUE: Portable semi-upright chest x-ray. There are no prior studies for comparison. There is an endotracheal tube with the tip approximately 6 cm above the carina. There is cardiomegaly. The pulmonary vessels are indistinct. The mediastinum is unremarkable. There are multiple bilateral pulmonary infiltrates slightly greater in the perihilar region. There is no pneumothorax, there are no large effusions. There are degenerative changes throughout the spine. IMPRESSION: CardiomegalyFINAL REPORT HISTORY: Line placement. A right IJ line has been inserted terminating in the upper SVC. There is no other change in the chest since the last chest xray at 11:27 PM on [**2138-8-2**]. Extensive bilateral fairly symmetric air space disease is noted. The heart is enlarged. An opaque catheter overlies the course of the proximal esophagus which may be an NGT. It cannot be seen distally below the level of the thoracic inlet. IMPRESSION: Right IJ line insertion.FINAL REPORT INDICATIONS: SOB. CT SCAN OF THE CHEST USING A CT PA PROTOCOL: TECHNIQUE: Helically acquired contiguous axial images of the chest with IV contrast using a CT PA protocol. CONTRAST: 100 cc Optiray IV due to fast bolus and allergies. FINDINGS: There is no pulmonary embolism. There is bibasilar consolidation. There is a small left pleural effusion. There is no pericardial effusion. The tracheal bronchial tree is patent. There are no pathologically enlarged, hilar, mediastinal or axillary lymph nodes. There is diffuse regions of ground glass opacity involving both lungs. There are small regions of geographic sparring, predominantly in the periphery of the lungs. No pulmonary nodules or masses are identified. Incidental note is made of a nasogastric tube and endotracheal tubes. IMPRESSION: 1. There is no pulmonary embolism. 2. Bibasilar dense consolidation. Diffuse ground glass opacity involving the

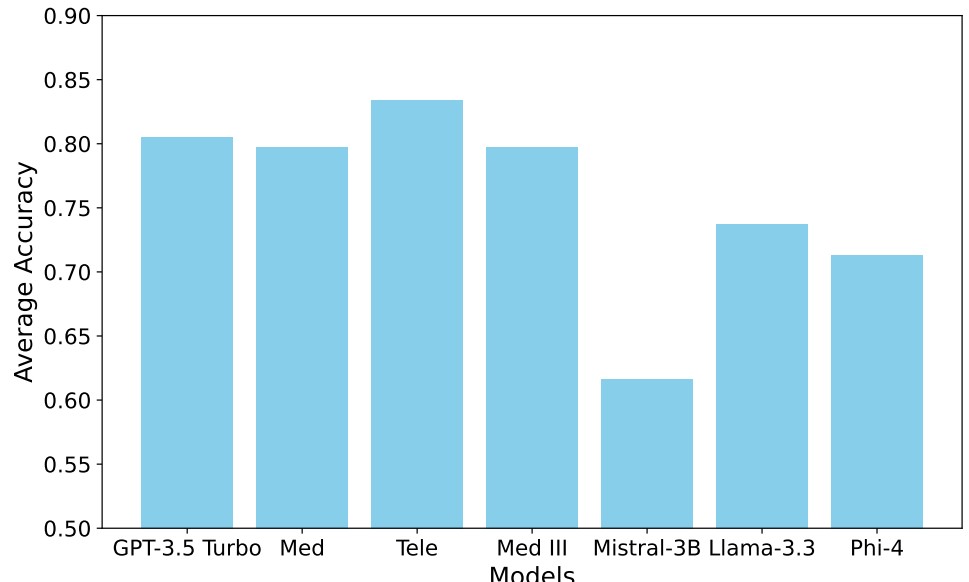

Figure 6: Average accuracies of LLMs (Med: GPT 4o finetuned on medical data, Tele: GPT 4o finetuned on telecommunications questions, Med III: GPT 4o finetuned on MIMIC III dataset) of different LLMs on the multiple choice telecommunications questions answering task.

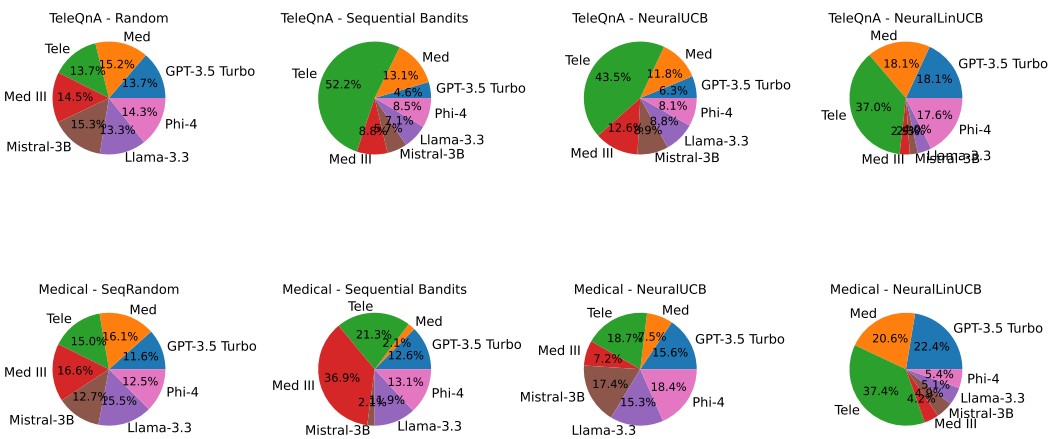

Figure 7: Model selection distribution for TeleQnA (1 task) and Medical (2 subtask) datasets (for diagnosis subtask) across different algorithms. Each pie chart shows the proportion of model usage.

lungs as described above. The differential diagnosis would include edema and infectious/ inflammatory etiologies. 3. There is no pericardial effusions and there is no large pleural effusion.FINAL REPORT CHEST SINGLE AP FILM: To reevaluate after diuresis. The endotracheal tube is 7 cm above carina. Right jugular CV line is in right brachiocephalic vein. Chest is rotated to the right. Probable cardiomegaly. Since the prior study there has been persistent ill-defined hazy opacity at the lung bases. No pneumothorax.FINAL REPORT AP SUPINE ABDOMINAL RADIOGRAPH INDICATION: Tympanic abdomen on exam, respiratory failure of uncertain etiology. There are multiple dilated loops of small bowel overlying the abdomen which measure up to 5.7 cm. There is a small amount of air in the recturm. There are no abnormal intra abdominal or intra pelvic calcifications. The osseous structures are unremarkable. IMPRESSION: Multiple dilated loops of small bowel are concerning for mechanical obstruction probably of the distal small bowel.FINAL REPORT HISTORY: Assess change in lines and tubes. Single view of the chest is compared to a

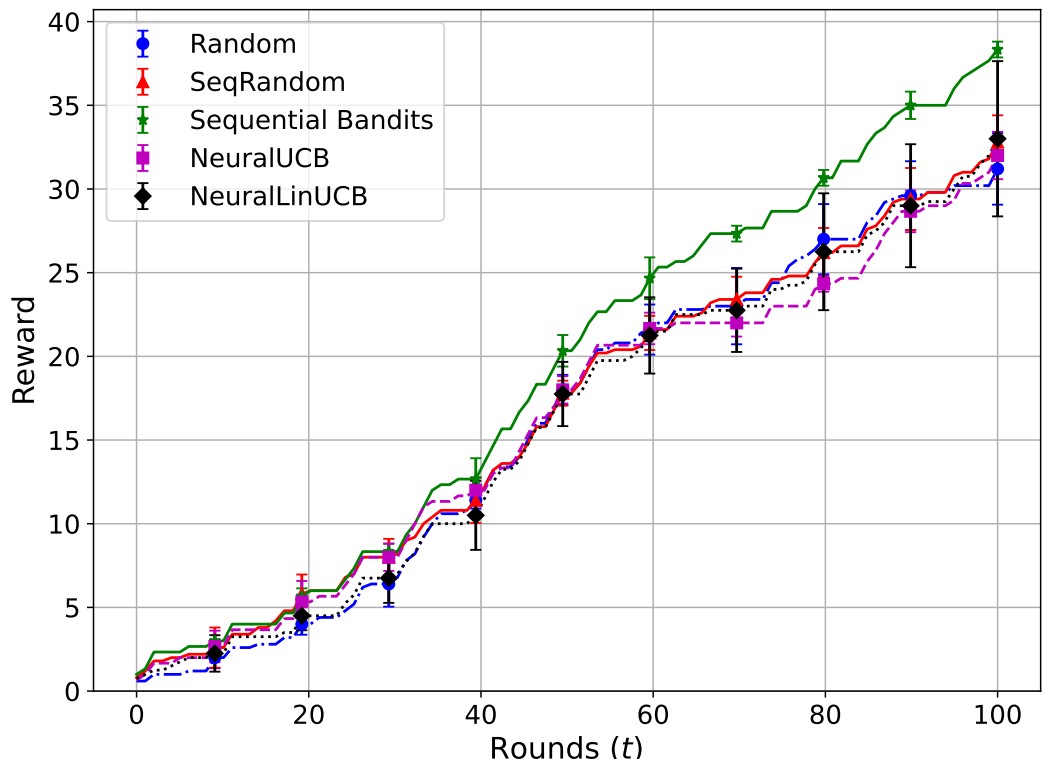

Figure 8: Cumulative reward for 2 subtask medical cost agnostic setting

previous study dated [**2138-8-5**]. The ETT is well above the carina. The right IJ line is in the right brachiocephalic vein. The cardiomediastinal contours are stable. There are persistent hazy opacities at both bases. There is no pneumothorax. IMPRESSION: Hazy illdefined opacities at both bases which may represent atelectasis although infection cannot be excluded.FINAL REPORT INDICATIONS: Decreased breath sounds on the left. PORTABLE CHEST: Comparison is made to previous films from [**2138-8-8**]. The ET tube and right IJ line are unchanged in position. The tip of the NG tube is not identified. The cardiomediastinal contours are stable. There is bilateral basilar hazy opacity. There is no pneumothorax. IMPRESSION: Increasing bilateral pleural effusion and associated atelectasis.FINAL REPORT IMPRESSION: Resolution of lower lobe opacities. FINDINGS: Portable AP chest radiograph is reviewed and compared to previous study of [**2138-8-7**]. The previously identified opacities in both lower lobes have markedly improved. These findings most likely represent improving aspiration pneumonia. The heart is mildly enlarged. The tip of the endotracheal tube is identified at the thoracic inlet. A nasogastric tube courses towards the stomach. No pneumothorax is seen.FINAL REPORT Fifty year old female with recent episode requiring intubation. Now post extubation. Shortness of breath. Assess for ischemia. SUMMARY OF EXERCISE DATA FROM THE REPORT OF THE EXERCISE LAB: Persantine was infused intravenously for approximately 4 minutes at a dose of approximately 0.142 mg/kg/min with no reported anginal symptoms or ST segment changes. INTERPRETATION: One to three minutes after the cessation of infusion, MIBI was administered IV. Image Protocol: Gated SPECT. Resting perfusion images were obtained with thallium. Tracer was injected 15 minutes prior to obtaining the resting images. Stress and resting perfusion images show no evidence of myocardial perfusion defects. Noted is a prominent right ventricle. Ejection fraction calculated from gated wall motion images obtained after Persantine administration shows a left ventricular ejection fraction of approximately 55%. There are no wall motion abnormalities. IMPRESSION: No evidence of myocardial perfusion defects. Prominent right ventricle.

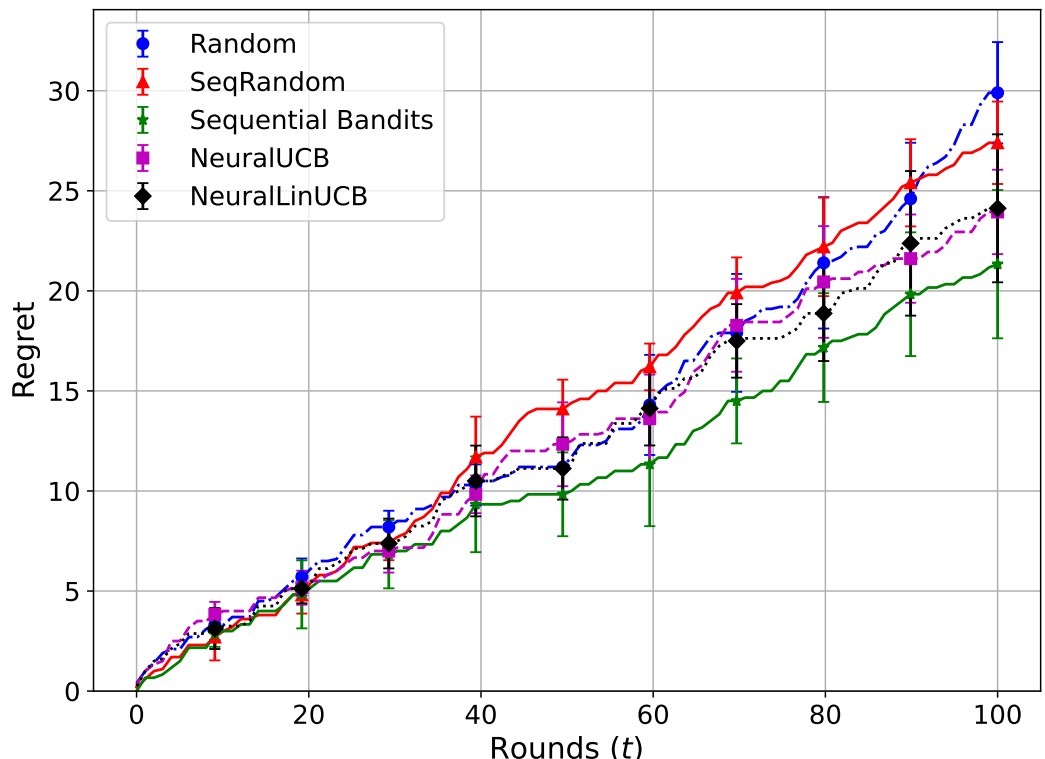

Figure 9: Cumulative regret for 2 subtask medical cost agnostic setting

We created the medical report above by combining multiple reports for the patient after different entries were made (after new observations, test results, imaging etc.) and removing any explicit mentions of the diagnoses. The diagnoses for the patient with the medical report above is pulmonary edema. Below we attach the summarized version of this report after being passed through an Assistant summarizer:

This extensive medical report describes multiple imaging studies performed on a fifty-year-old female patient with a history of recent intubation. The report includes findings from several chest X-rays, a CT scan of the chest, an abdominal radiograph, and results from a cardiac exercise lab.1. **Chest Imaging Findings:** - Multiple portable chest X-rays reveal cardiomegaly (enlarged heart) and bilateral pulmonary infiltrates predominantly in the perihilar region. Throughout the studies, there is persistent hazy opacity at the lung bases, which may represent atelectasis or infection. Cardiomegaly and stable cardiomediastinal contours are consistently noted. - The CT scan of the chest shows bibasilar consolidation, diffuse ground glass opacity potentially due to edema, and no pulmonary embolism. There is a small left pleural effusion without pericardial effusion. - The most recent chest radiograph shows resolution of lower lobe opacities, suggesting improvement in previously suspected aspiration pneumonia.2. **Abdominal Imaging Findings:** - The abdominal radiograph indicates multiple dilated loops of small bowel suggesting possible mechanical obstruction in the distal small bowel. There is no abnormal calcification.3. **Cardiac Exercise Lab Findings:** - Perfusion images show no evidence of myocardial perfusion defects, and ejection fraction is calculated at approximately 55%, with no wall motion abnormalities. Notably, there is a prominent right ventricle.4. **Line Placement and Tube Position:** - There are placements of an endotracheal tube, right internal jugular line terminating in the upper SVC or right brachiocephalic vein, and an opaque catheter possibly representing an NG (nasogastric) tube whose tip isn't identified. The positions are stable in subsequent images.Overall, the patient presents with cardiomegaly, resolving pulmonary opacities, possible small bowel obstruction, and a prominent right ventricle noted on cardiac imaging. There is no evidence of myocardial perfusion defects or pulmonary embolism, and the pleural

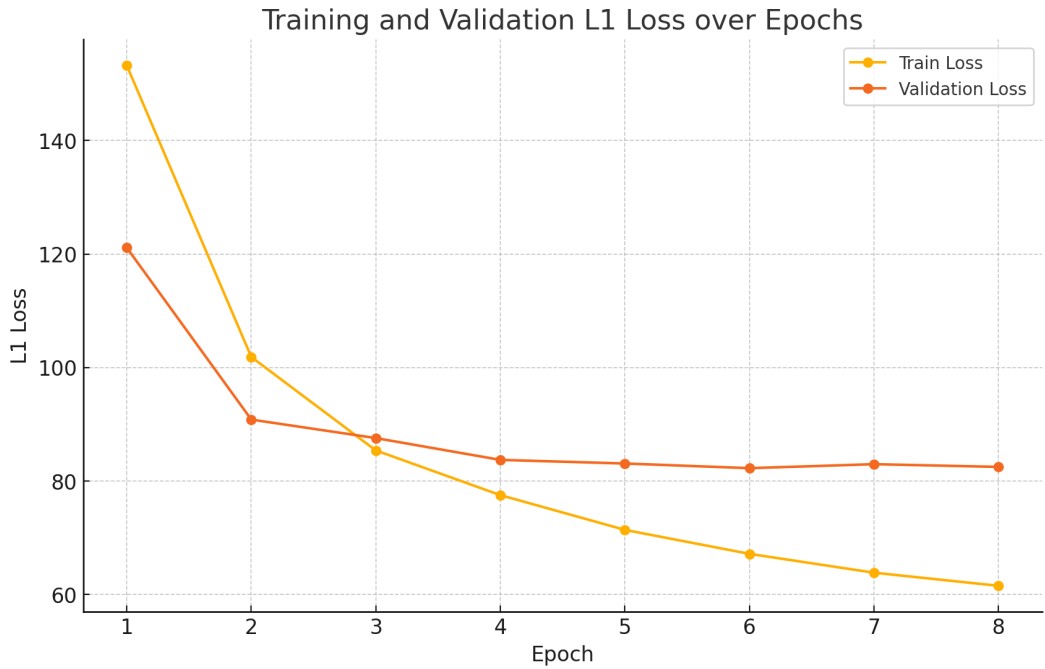

Figure 10: Training and validation loss over epochs for Bert regression model for output token length prediction

effusion is small.Provide a medical diagnosis based on the provided medical report summary above. This is for experimentation purposes only.

As seen above, the summarization of the medical report above highlights the most important information that may be helpful in making a diagnosis and hence simplifies the task for the next LLM in the pipeline.

### B.3 OUTPUT TOKEN LENGTH PREDICTION MODEL

Next, we give more details on the output token length prediction model that we use to predict the number of output tokens given the input prompt length. As we briefly explained in the main paper, we train a Bert regression model with L1 loss to train this model using the LMSYS-Chat 1M dataset Zheng et al. (2023).

We present the training and validation loss curves over the number of epochs for the Bert regression model with L1 loss using 50,000 examples with a 90% to 10% training and validation split in Figure 10. The validation loss starts to increase after 6 epochs, hence we used the model trained up to 6 epochs for output token length prediction in our experiments as the model we use before making online updates to this model in our algorithm. This model is only trained once before starting the experiments for cost aware settings and is iteratively updated by taking an AdamW optimizer step every 5 token length observations with a learning rate of $10^{-6}$ during the training process shown in Algorithm 1. We will now provide the absolute average cost estimation error over all rounds for the different models that we use in our experiments. Figure 11 shows the cost estimation error for the different models for the offline token length prediction model. It can be seen that Med III has considerably higher cost estimation error compared to the other models, which might be due to the fact that it was finetuned on a highly specialized dataset, making it difficult to predict. Figure 12 shows the mean absolute error for the token length estimation for the online cost prediction model across all subtasks for the telecom experimental setting. Comparing Figure 11 to Figure 12 shows that the online updates especially benefits the finetuned models Med, Tele and Med III. This is expected, as finetuned models tend to give less verbose, predictable output lengths compared to other

models like GPT or Llama. Llama seems to obtain relatively high estimation error according to both plots, a possible explanation for this is that the offline training for the cost model before starting the online training was done with the Llama 13B model as the LMSYS Chat 1M includes examples from this Llama model while the API calls we are making are for the Llama 70B model. This may explain why even after online training the error for Llama is higher compared to other models. Similarly for the Assistant, the experiments in Figure 11 were done when the Assistant model was hosted on GPT 3.5-Turbo, but Azure was no longer hosting that model when we conducted the experiments in Figure 12 and GPT 3.5-Turbo was replaced with GPT 4.1 mini. Also, LMSYS Chat 1M dataset had datapoints for GPT 3.5-Turbo which might have again caused a mismatch like in Llama during the offline training phase, contributing to higher error. Finally, we look at the variation of MAE and RMSE over time when we train the token length predictor online as shown in Figure 13. Both MAE and RMSE show significant decreases in error, highlighting the benefit of online training. Every evaluation step corresponds to taking an optimizer step after observing 5 input token length-output token length pairs from the chosen models. In our experimental settings, we initialized the cost model at the beginning of the first run for each of the baselines with the offline trained model and train the cost model throughout all the runs so that the next run uses the improved model from the previous run. However, to ensure fairness across the different baselines, we did not use the online trained cost model across different baselines as this would create an advantage for the baselines that are run later on since their cost prediction model would be more accurate.

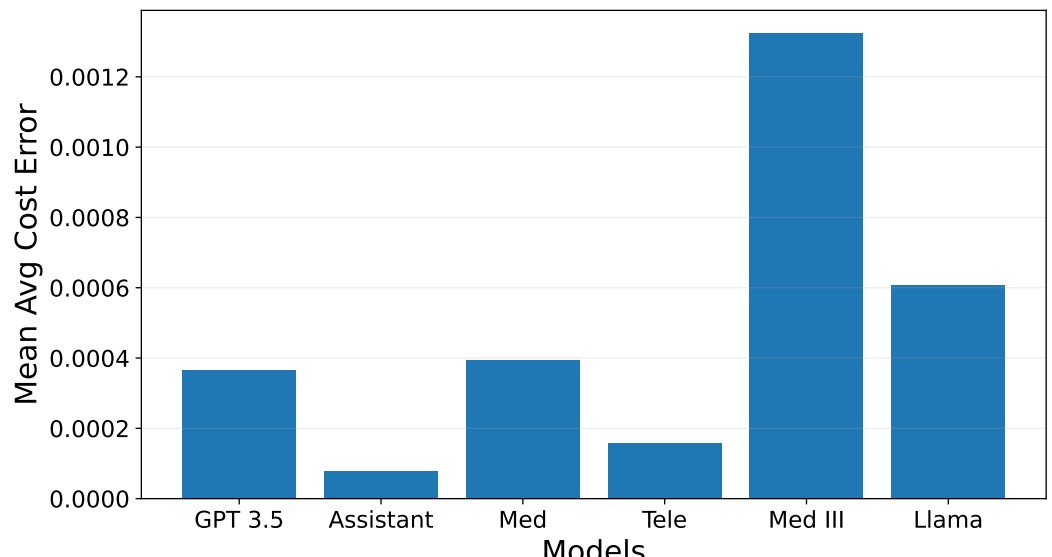

Figure 11: Average cost estimation error for the different models with the offline token length prediction model

## B.4 CALCULATION OF LLM DEPLOYMENT COSTS

To construct the total cost plots that can be seen in Figure 3 we use the per token pricing information that is available on Microsoft Azure, which is where we run all our experiments. The cost per input token for GPT 3.5 Turbo is $0.0000005, for GPT 4.1-mini is $0.0000001, $0.00000025 for finetuned GPT 4o models (Med, Tele, Med III) and $0.00000071 for Llama. The cost per output token for GPT 3.5 Turbo is $0.0000015, for GPT 4.1-mini is $0.000004, $0.00001 for finetuned GPT 4o models (Med, Tele, Med III) and $0.00000071 for Llama. Hence the finetuned models are the most expensive models, then Llama and the cheapest is GPT 3.5 Turbo. We use these per token costs for input and output and multiply them with the actual number of input tokens and the predicted number of output tokens to get the corresponding estimated cost values for using these LLMs. To calculate the actual costs, we use the actual number of output tokens instead, after observing the outputs of the LLMs.

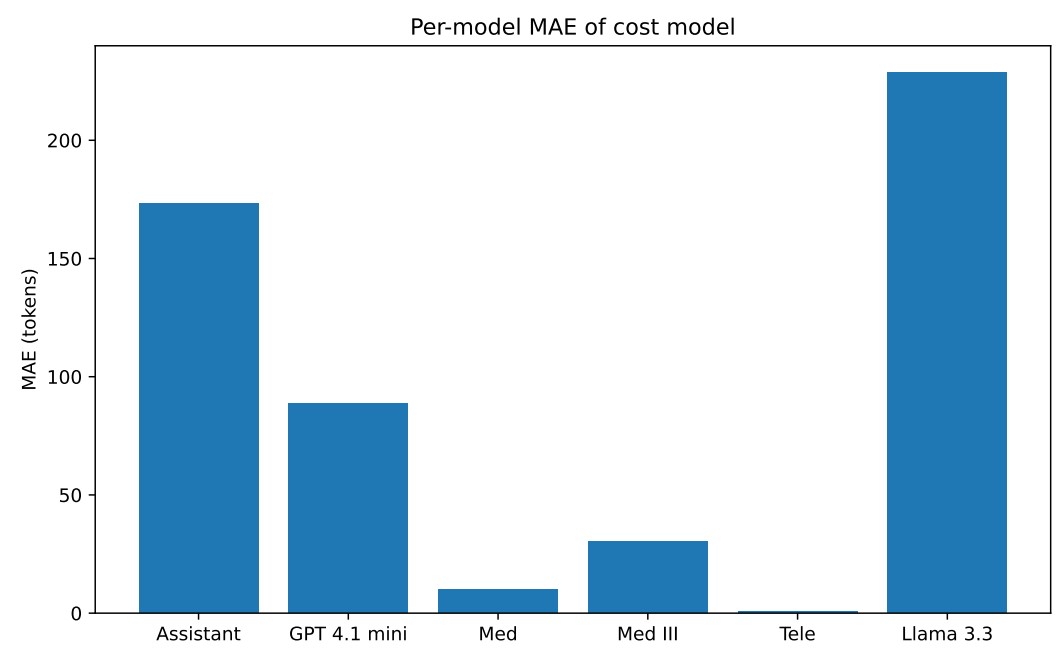

Figure 12: Mean absolute token length estimation error for the different models with the online token length prediction model

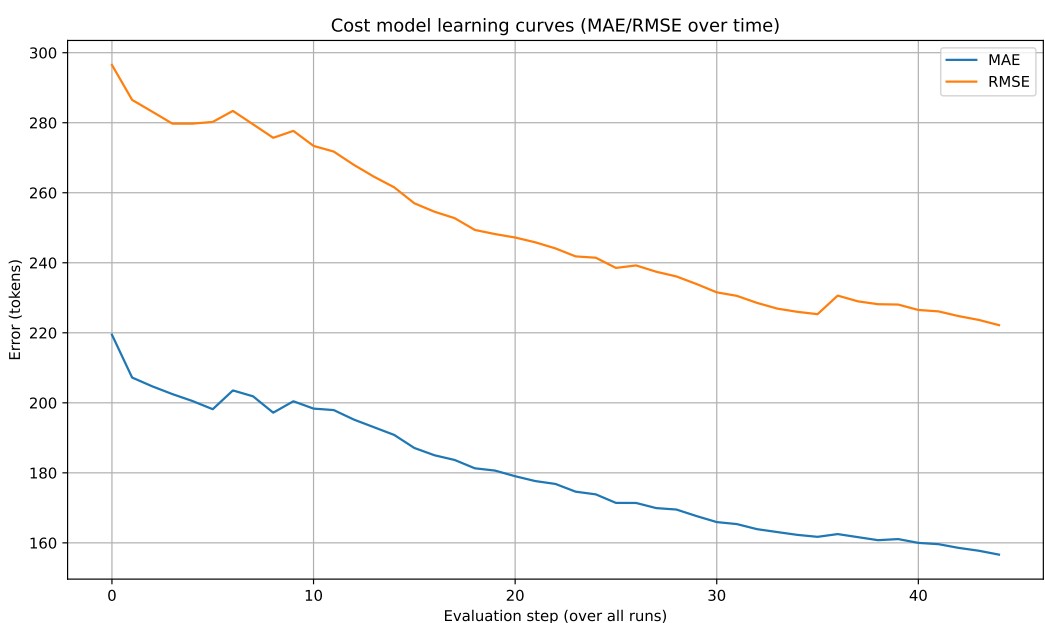

Figure 13: MAE and RMSE variation with the evaluation steps (model updates) for the token length prediction model

## C    COST AWARE SETTING RESULTS

In this section we present extra results for the 3 subtask telecommunication setting we detailed in the main paper such as the model selections with pie charts, accuracy of models for this setting as well

as results for the 3 subtask version of the medical pipeline we considered before and extra results for the 2 subtask medical pipeline we detail in the main paper. We want to note that the cumulative regret results we present here are in terms of the regret for the reward rather than net reward, hence the presented regret plots do not take cost into consideration when computing regret.

## C.1    3 SUBTASK MEDICAL SETTING

We consider a task decomposition into three sequential subtasks: summarization, debate, and diagnosis. The rewards for the debate subtask is obtained by getting a pro-hallucination and a con-hallucination bias on the summary from before and getting the difference. The results presented here are averaged over 5 independent runs.

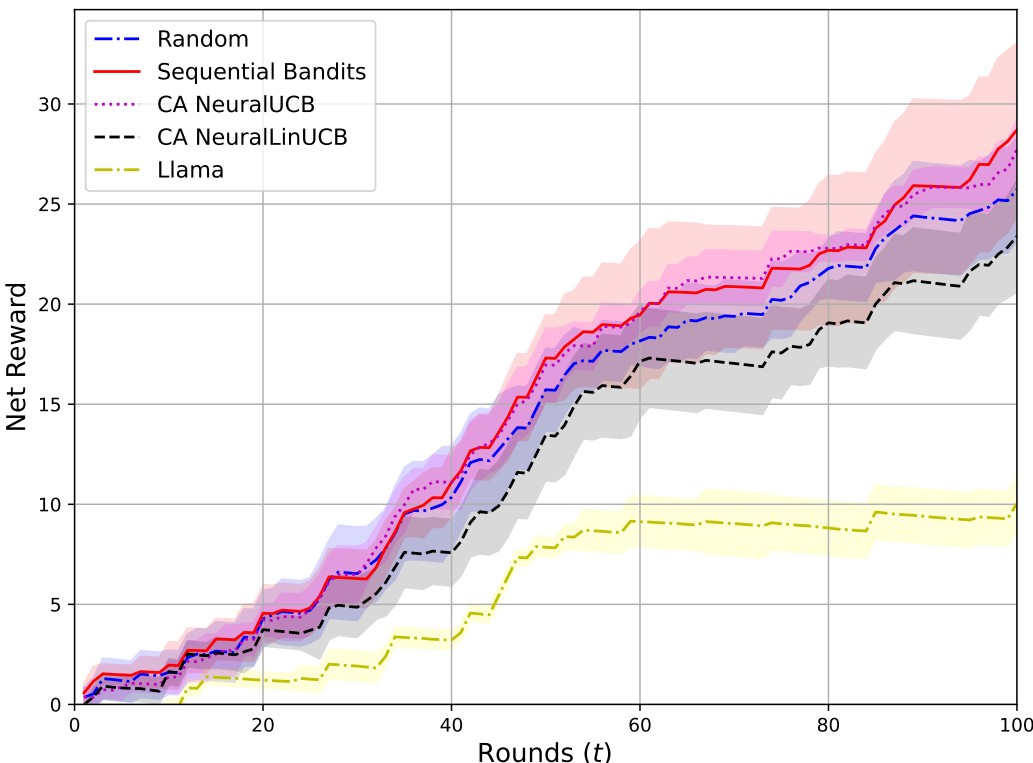

Figure 14: Net reward for 3 subtask medical setting

As can be seen in Figure 14, our algorithm indicated by the solid red line achieves the highest net reward in this cost aware setting. Cost Aware NeuralUCB also performs comparably to our algorithm, indicating its success. Random surprisingly outperforms both Cost Aware NeuralLinUCB and Llama and Llama performs considerably worse then all the other algorithms which indicates that static LLM selection strategies in these more complex pipelines might not be very reliable and effective.

Now, we consider the regrets of the algorithms as shown in Figure 15. It can be seen that the regret of the algorithms are mostly similar, except for Llama which performs considerably worse. There is some randomness involved with the regret as we measure it as the difference in the performance of the best model for that round compared to the performance of the selected model. This means that even if we are able to identify the best model for this subtask in hindsight, we may still incur regret as that model may perform poorly for a specific type of input whereas another gets an accuracy of 1, making us incur regret. Hence, this can be considered as a more competitive regret than is usually measured.

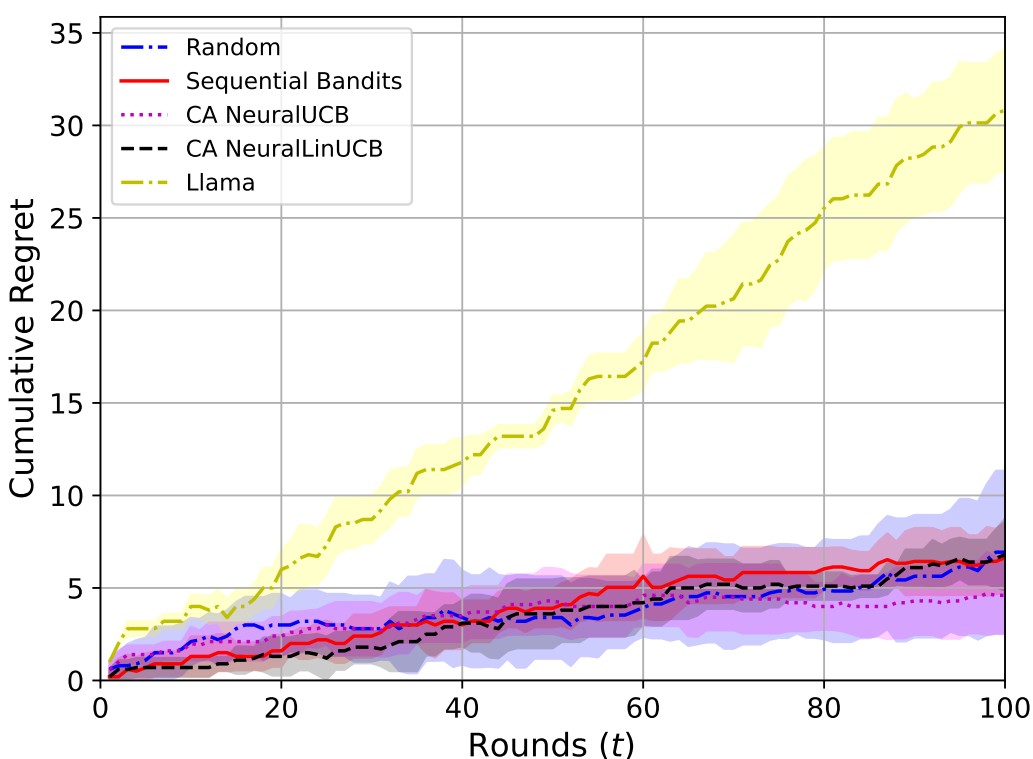

Figure 15: Cumulative regret for 3 subtask medical setting

Next, we show the total cost incurred by the algorithms for the 3 subtask medical setting. As can be seen from Figure 16, Cost Aware NeuralLinUCB gets the highest cost, followed by Random, Llama, Sequential Bandits and Cost Aware NeuralLinUCB. It can be observed that the Llama baseline gets the lowest standard deviation when compared to the other algorithms for total cost as its shaded regions are barely visible. This is an advantage of the static LLM selection baselines as the total cost that they get show a low variation among different runs.

Next, we show the average accuracy of the models for the diagnosis subtask in this 3 subtask pipeline setting as shown in Figure 17. As in Figure 1, here accuracy is an indication of how many diagnoses the LLM was able to predict correctly. It can be seen that Llama obtains the highest mean accuracy followed by GPT 3.5 Turbo, Tele, Med III and Med which is a similar ordering to the one we had in Figure 1.

For the 3 subtask medical setting, the first row of Figure 22, shows the model selections for the diagnosis subtask. Sequential Bandits and CA NeuralUCB are the best performers as is evident from their model choices, as they select Med less frequently compared to the others and select Llama more.

## C.2    3 Subtask Telecom Setting

In this section, we give more results for the cost aware 3 subtask telecom setting, which we introduced in the main paper. First, we present the cumulative regret for this setting as can be seen in Figure 18. We observe a similar trend to the net reward in Figure 3 where Sequential Bandits outperforms the other baselines also in terms of regret and CA NeuralUCB performs the worst, while CA NeuralLinUCB and Random's performances are quite similar.

Next, we look at the average accuracy of the models across all the subtasks. First we look at the summarization subtask, whose results can be seen in Figure 19 Med, Tele and Med III achieve

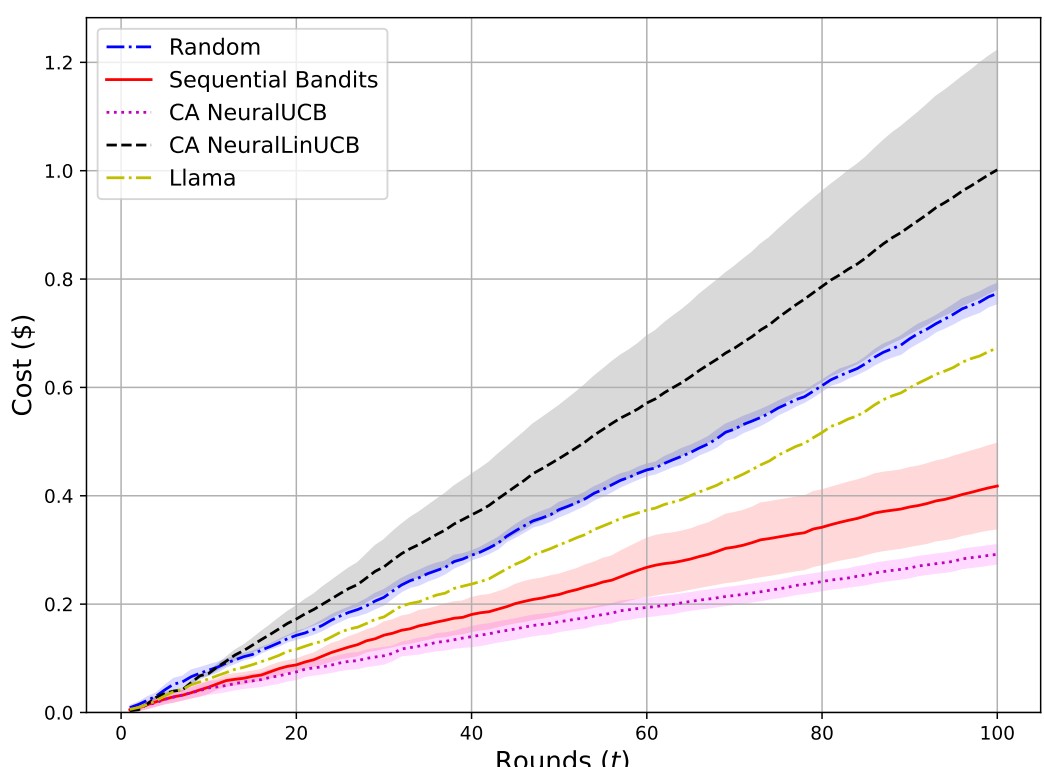

Figure 16: Total cost for 3 subtask medical setting

competitive and similar summarization rewards at $\approx \%95$, while Llama achieves around $\%80$ and Assistant does considerably worse at $\%2$. For the answering subtask, seen in Figure 20, all models do relatively similarly, with Tele doing the best among them, as expected. For the explanation subtask, seen in Figure 21, the Med III model performs the best followed by Llama and Med models.

Now, we analyze the model selections each of the algorithms made for summarization and answering subtasks. Figure 23, 24 show the model selections for the summarization and answering subtasks. For Sequential Bandits we see that Llama is selected the most often followed by GPT-4.1 mini while for CA NeuralUCB it selects nearly evenly between Assistants and GPT-4.1 mini, showing its suboptimality as Assistants gets the lowest summarization accuracy by far. CA NeuralLinUCB similarly shows near uniform selection as it did for the explanation subtask while CA NeUCB Joint seems to prefer Llama, GPT 4.1 mini and Assistants. For the answering subtasks, Sequential Bandits exhibits a near uniform selection, which makes sense given that the gap in accuracy between the models is quite small for this subtask. Similarly, CA NeUCB Joint also shows near uniform selection while CA NeuralUCB and CA NeuralLinUCB prefer GPT-4.1 mini, Llama and Med, Llama the most respectively.

## C.3    2 SUBTASK MEDICAL SETTING

In this section, we present some additional results for the 2 subtask medical setting which we talked in detail in the main paper.

First, we present a result on the average accuracy of the models for the summarization subtask for 10 runs as shown in Figure 25. Assistant is a custom specialized GPT 3.5 Turbo model that we used instead of Med III, as using the Med III model for summarization triggered some errors related to safety from OpenAI's policy. The maximum available accuracy for the summarization task is 100 and it can be seen that GPT 3.5, Tele and Llama perform very similarly and are the best summarizers while Assistant and Med perform worst.

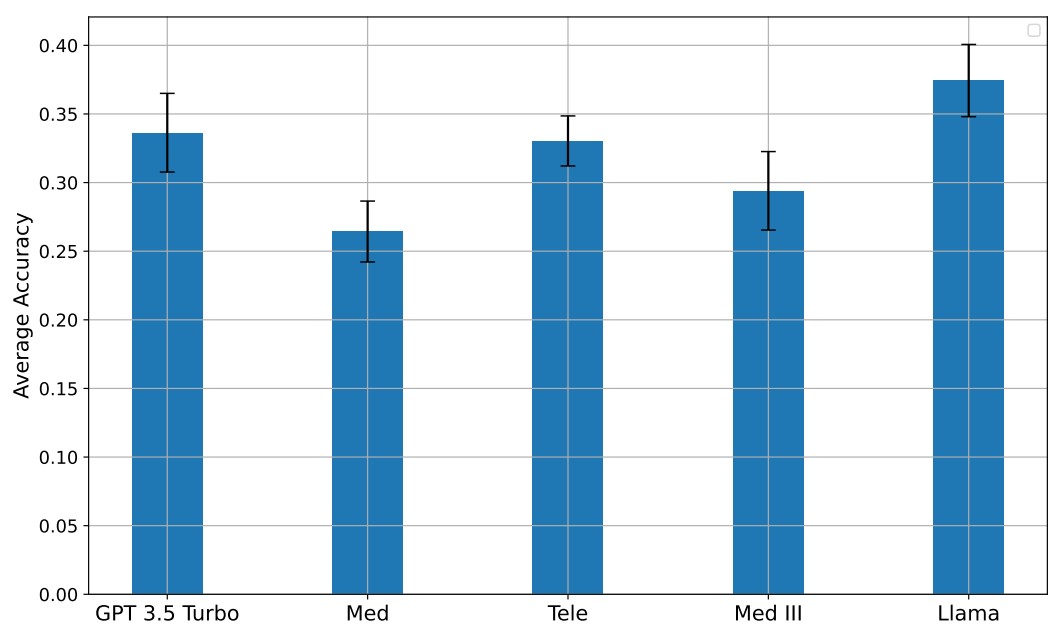

Figure 17: Average accuracy of models for the diagnosis subtask in 3 subtask medical setting given the summarizer is selected at random

Next, we investigate whether a higher accuracy summary necessarily leads to higher accuracy for diagnosis. This result, present in Figure 26 shows that this is not necessarily the case, though there is some correlation between the two. Figure 26 essentially shows the mean diagnosis accuracy when a particular model is chosen as the summarizer. Hence, the values presented are an average of the diagnoses prediction resulting from the summaries given by the models shown. In this case, both the diagnoser and the summarizer are selected at random to reduce any bias. If a good summary consistently leads to a successful diagnosis, we would expect Figures 25 and 26 to look similar in terms of their ordering. While Assistant has the lowest accuracy, the rest of the models are quite similar in terms of accuracy as seen in Figure 26 while this is not the case in Figure 25. Moreover, the standard deviation values shown by the error bars in Figure 26 are quite large, indicating a high variance which makes it even harder to conclude that there is a direct correlation between summary accuracy and diagnosis accuracy. Therefore, this shows that it is not possible to prioritize accuracy early on in the pipeline by setting a low $\alpha$ for summarization and then prioritizing cost by setting a high $\alpha$ as a high accuracy summary doesn't necessarily guarantee a high accuracy for the diagnoses.

We now look at the average accuracy of the models for the diagnosis subtask in this cost aware setting. Figure 27 shows the accuracies of the models for this subtask with Llama and GPT 3.5 having the highest accuracies and Med the lowest.

Next, we present the cumulative regret results for this setting as shown in Figure 28. As expected, Random gets the highest regret while the rest of the algorithms perform quite comparably with Sequential Bandits and Llama baselines slightly outperforming CA NeuralUCB and NeuralLinUCB. This result is consistent with the net reward result for 2 subtask medical setting we presented in the main paper.

Finally, we look at the model selections for the summarization subtask in this setting. Figure 29 shows the model selections of all the algorithms. As mentioned in the main paper, even though Sequential Bandits and CA NeUCB Joint had similar selections for the diagnosis subtask, since their selections for the summarizers differ as shown, this causes the main performance difference. These selections also explain why CA NeuralLinUCB incurs the second highest cost among all algorithms, as it selects Tele very frequently (%59.4) which is the most expensive model along with the other finetuned models.

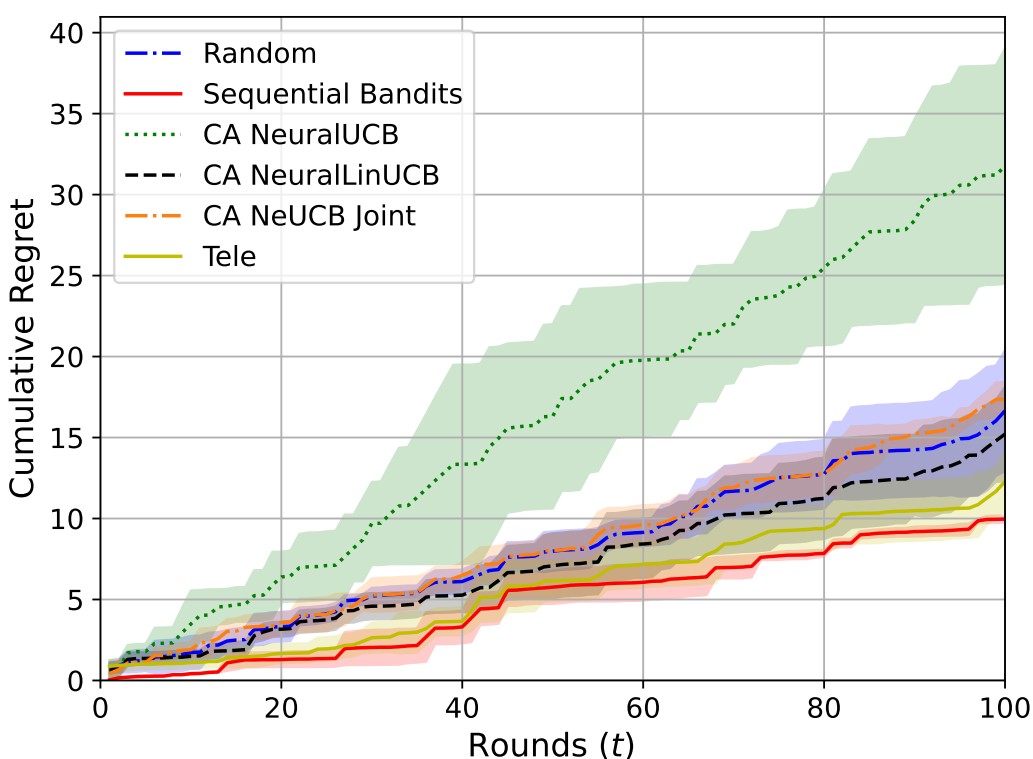

Figure 18: Cumulative regret for 3 subtask telecom setting

## D RESULTS ON RESPONSE LATENCY VS TOKEN LENGTH

Finally, we present some results on response latency and its variation with the token length. As we mentioned in the main paper, response latency could also be considered as a cost and this is a possible future direction of this work of incorporating response latency as well into the objective. We first start by running Random selection over 1000 instances (10 runs) for the summarization subtask for the medical dataset and do a scatter plot of all points as shown in Figure 30. We see that Assistant seems to have a higher latency overall when compared with the other models but there doesn't seem to be a specific trend showing that increasing total number of tokens leads to a higher response latency. We see a similar trend when we look at the response latency variation with the number of tokens for the diagnosis subtask in Figure 31 with the major difference being that the number of tokens are much lower compared to summarization. We notice some outliers specifically by the Llama model as shown by the yellow scatters in Figure 30. We expect this to be because there may be congestion on Llama as it is one of the most widely used LLMs.

Next, we show the average response latency with the total number of tokens for the different models. It can be seen from Figures 32 and 33 that as we could understand from Figure 30, Assistant has the highest average response latency even though it has the lowest total number of tokens. Med and Tele have similar latency values as they are finetuned from the same base model GPT 4o and Llama has the highest latency after Assistant. We see similar trends for the average response latency for the diagnosis subtask in Figure 33 as Llama has higher latency than the rest of the models, which show similar latencies.

Finally, we take a look at the scatter plots for response latency against number of tokens for some of the models separately so see if there is some correlation when we consider the models separately. Figures 34, 35 and 36 show these variations for Med III, GPT 3.5 Turbo and Llama models respectively for the diagnosis subtask. There are 500 points in each of these figures. While there does not seem to be a particular trend between response latency and number of tokens, it can be observed that

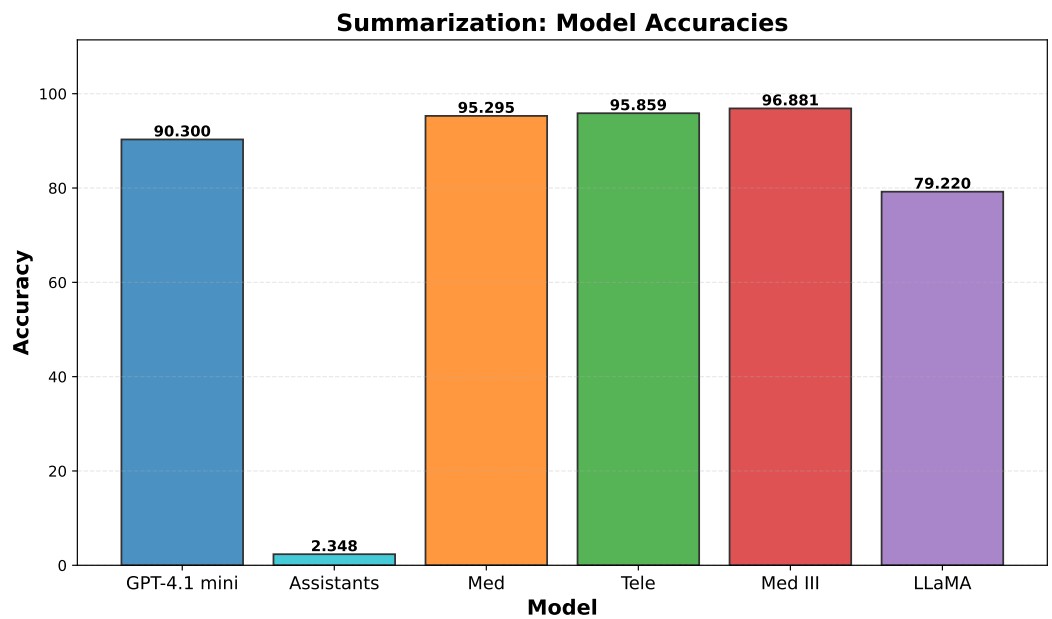

Figure 19: Average accuracy of models for the summarization subtask given the summarizer is selected at random

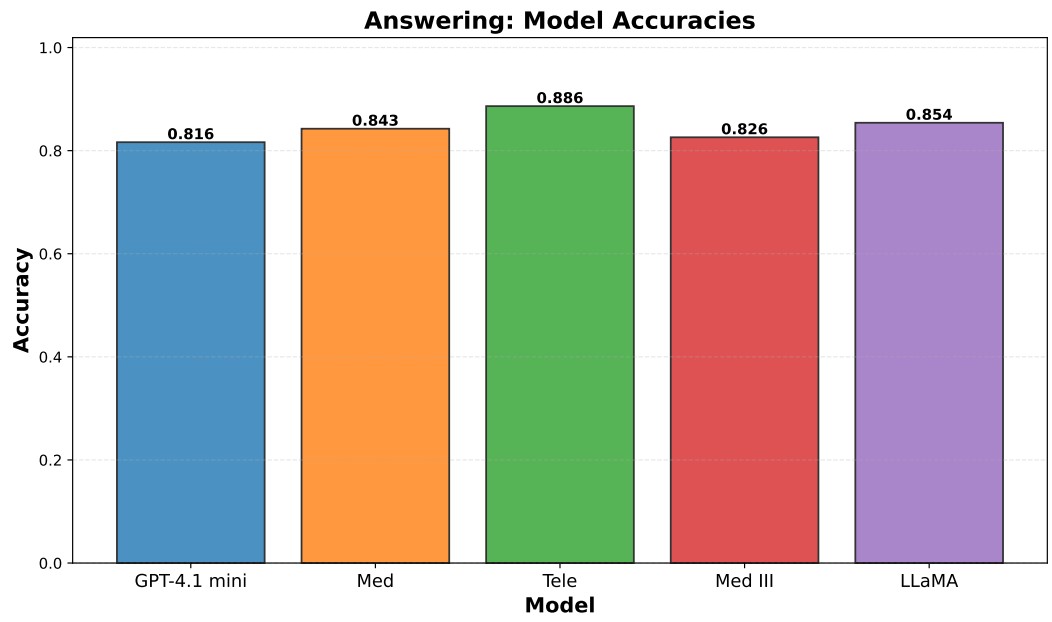

Figure 20: Average accuracy of models for the answering subtask given the summarizer and answerer are selected at random

the more utilized models through APIs such as Llama and GPT 3.5 show a lot more variation and increased latency times for some queries as indicated by the outliers in the plots which are numerous. On the other hand, Figure 34 shows that this is a much rarer occurence for the Med III model which is our own finetuned model, which illustrates the benefit of using finetuned models which are not widely used or available to others. This further motivates the need for LLM selection algorithms, as

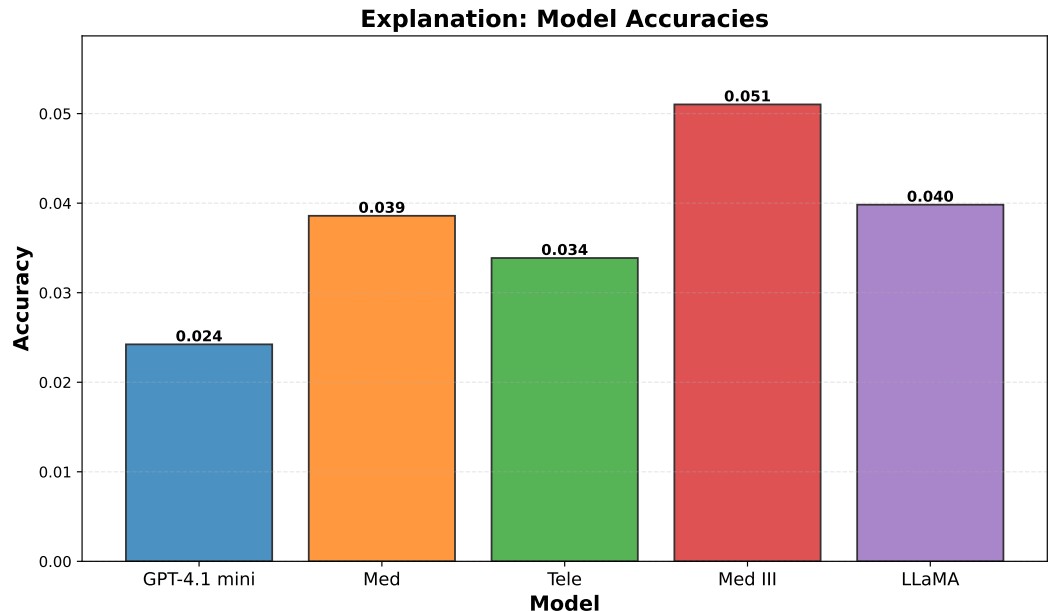

Figure 21: Average accuracy of models for the explanation subtask given the summarizer, answerer and explainer are selected at random

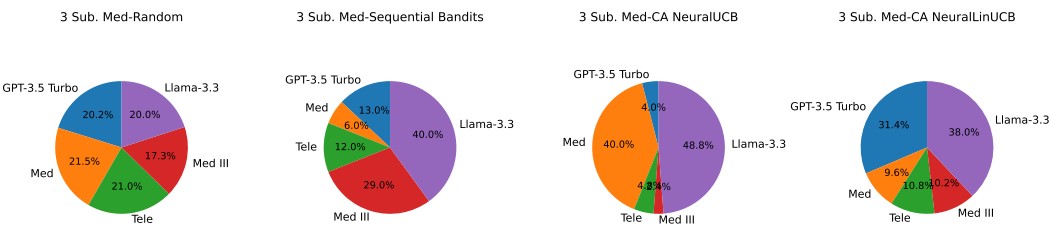

Figure 22: Model selection distribution for Medical (3 subtask) diagnosis subtask across different algorithms. Each pie chart shows the proportion of model usage.

we can avoid choosing the higher latency models if we can identify that certain specialized models for our task at hand can perform better or similarly but with a much improved response latency.

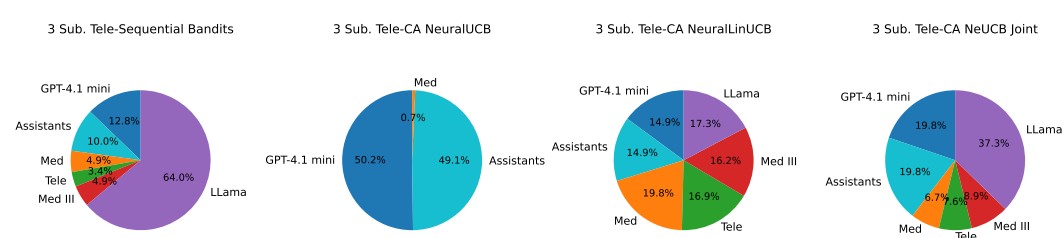

Figure 23: Model selection distribution for Telecom (3 subtask) summarization subtask across different algorithms. Each pie chart shows the proportion of model usage.

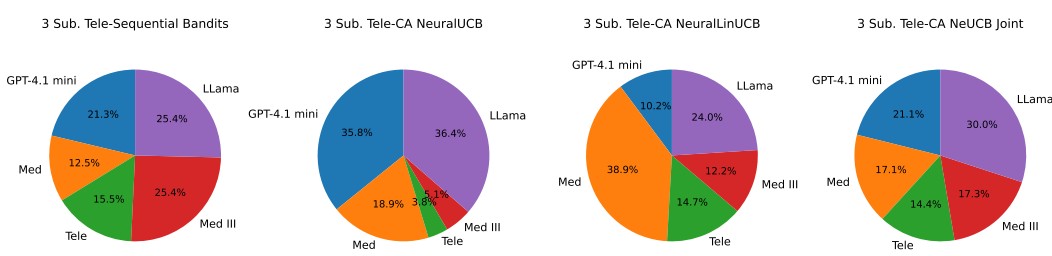

Figure 24: Model selection distribution for Telecom (3 subtask) answering subtask across different algorithms. Each pie chart shows the proportion of model usage.

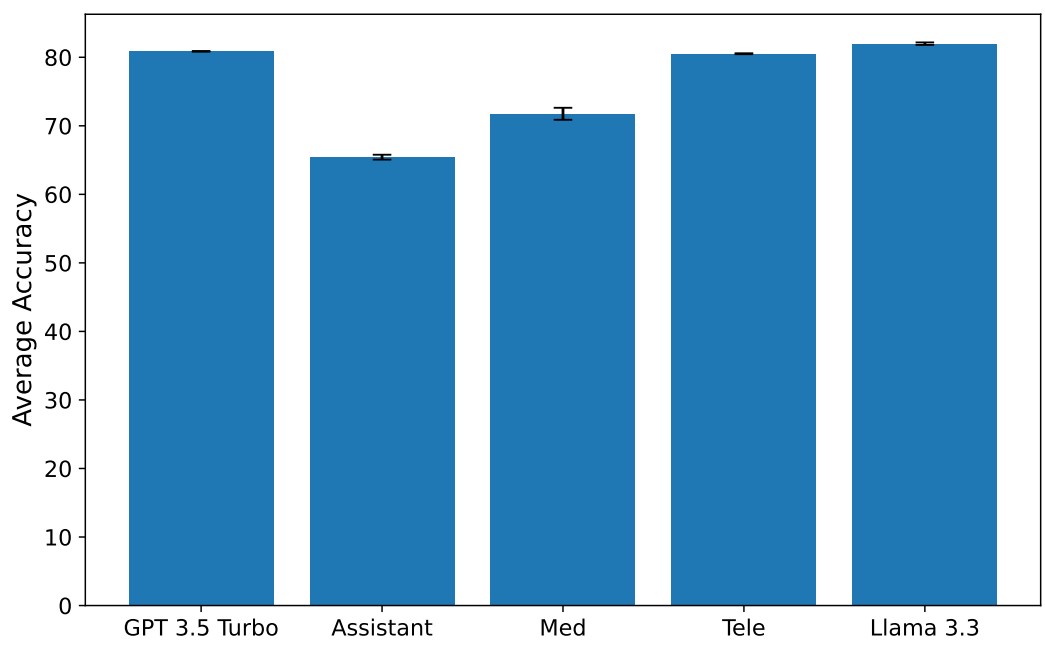

Figure 25: Average accuracy of models for the summarization subtask given the summarizer is selected at random

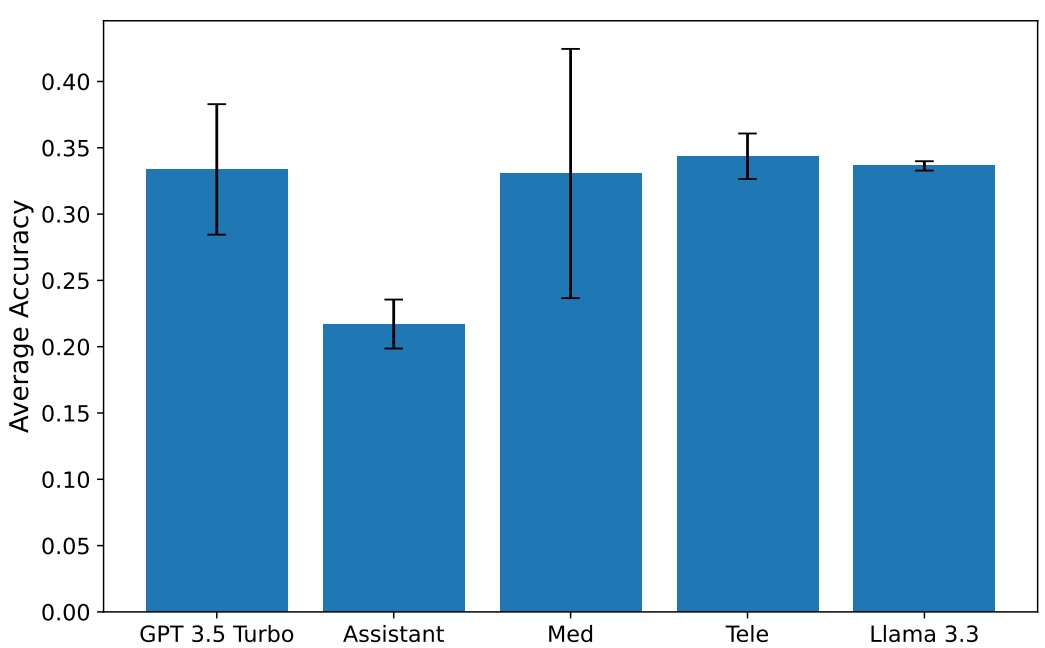

Figure 26: Diagnosis accuracies given the summarizer is selected as the model shown for random diagnoser selection

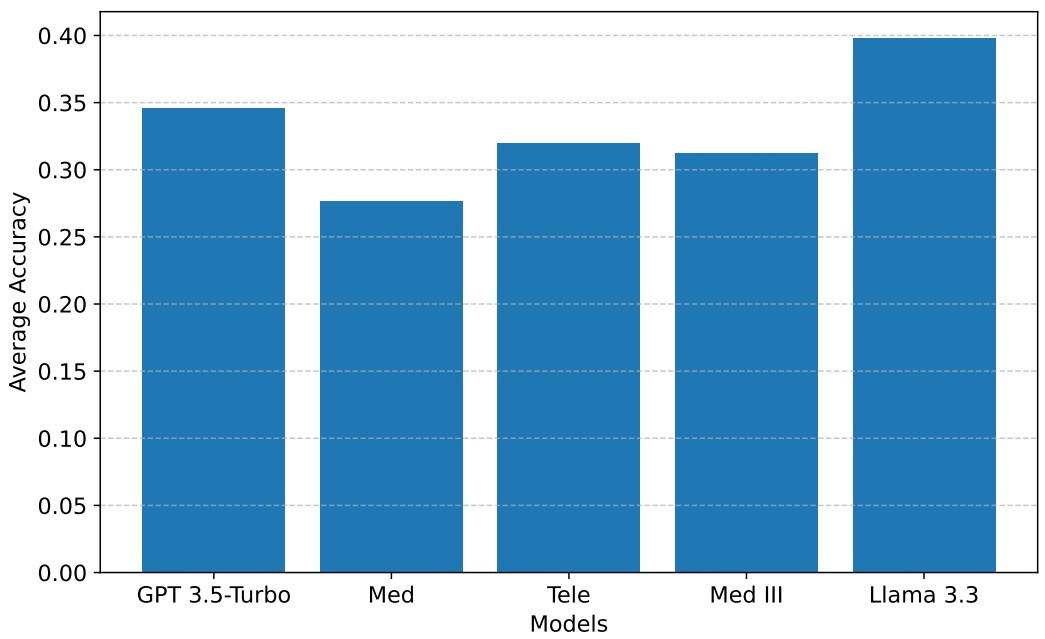

Figure 27: Average accuracy of models for the diagnosis subtask for random summarizer and diagnoser selection

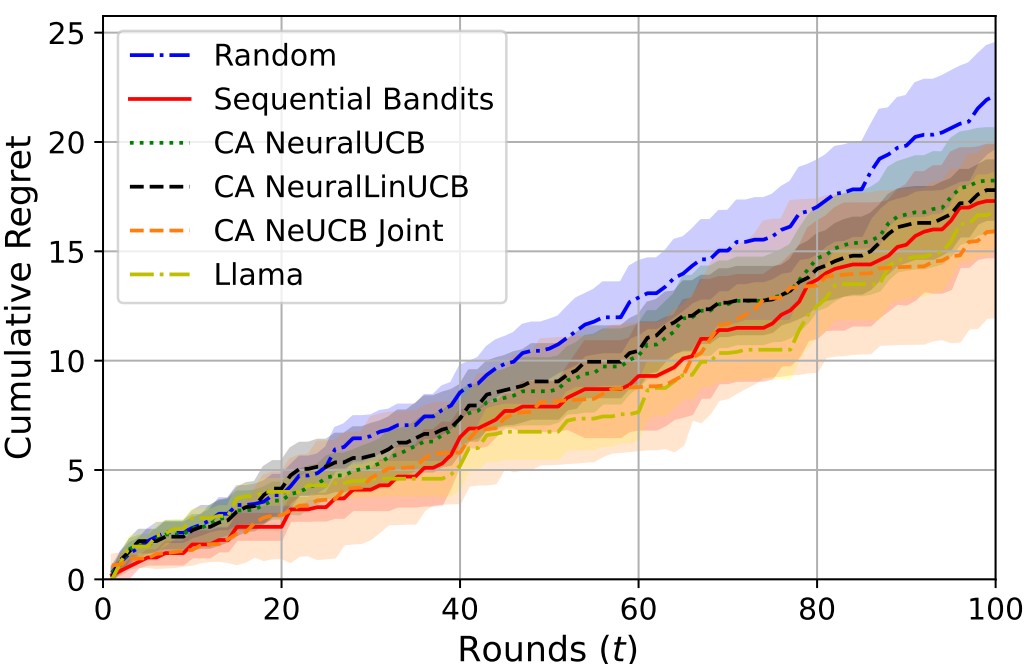

Figure 28: Cumulative regret for the diagnosis subtask in 2 subtask medical setting

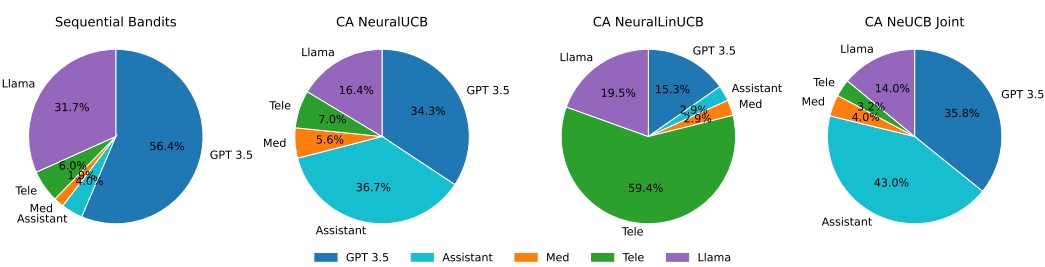

Figure 29: Model selection distribution for Medical (2 subtask) dataset summarization subtask across different algorithms. Each pie chart shows the proportion of model usage.

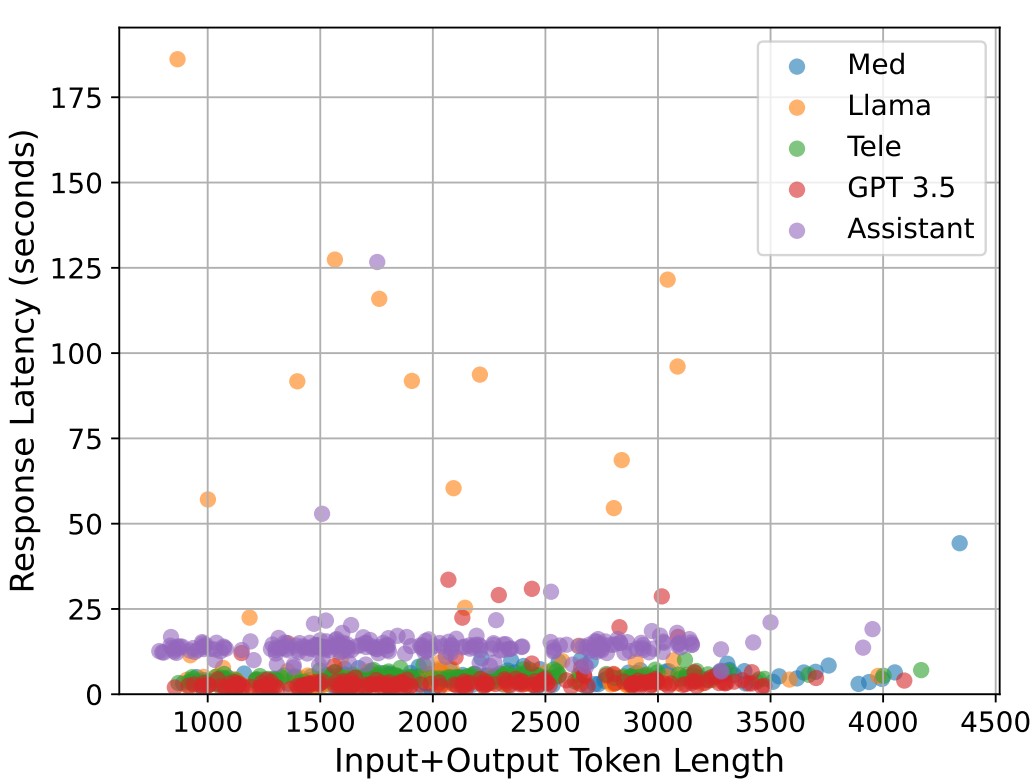

Figure 30: Scatter plot showing response latency against the sum of the number of input and output tokens for the summarization subtask

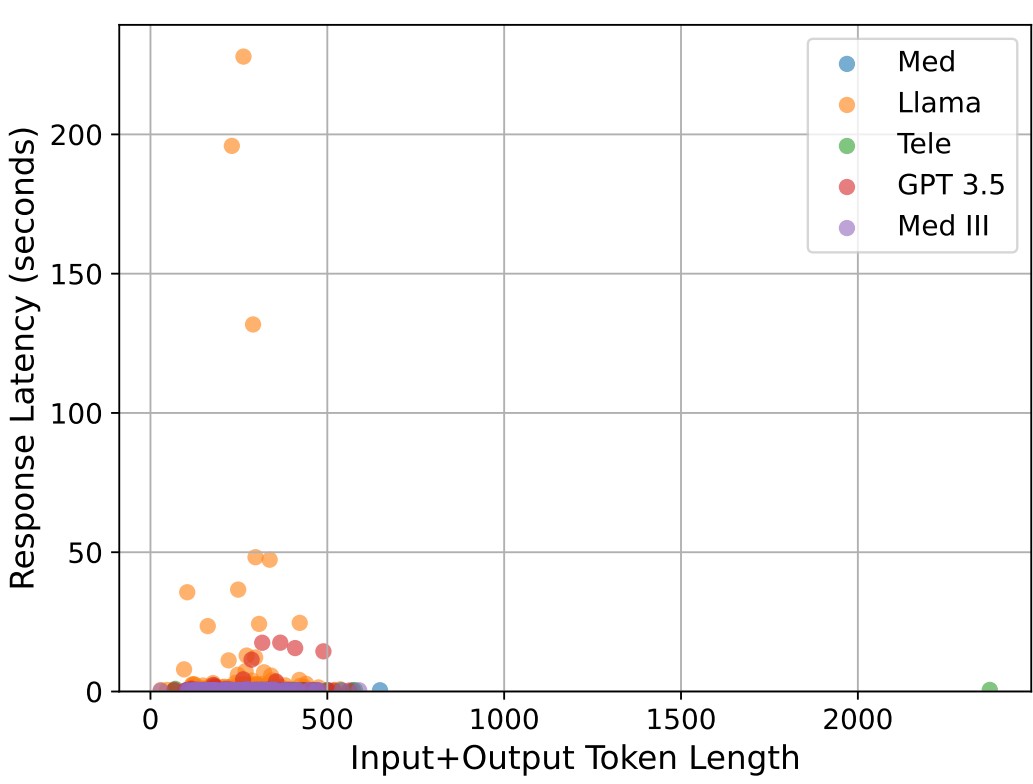

Figure 31: Scatter plot showing response latency against the sum of the number of input and output tokens for the diagnosis subtask

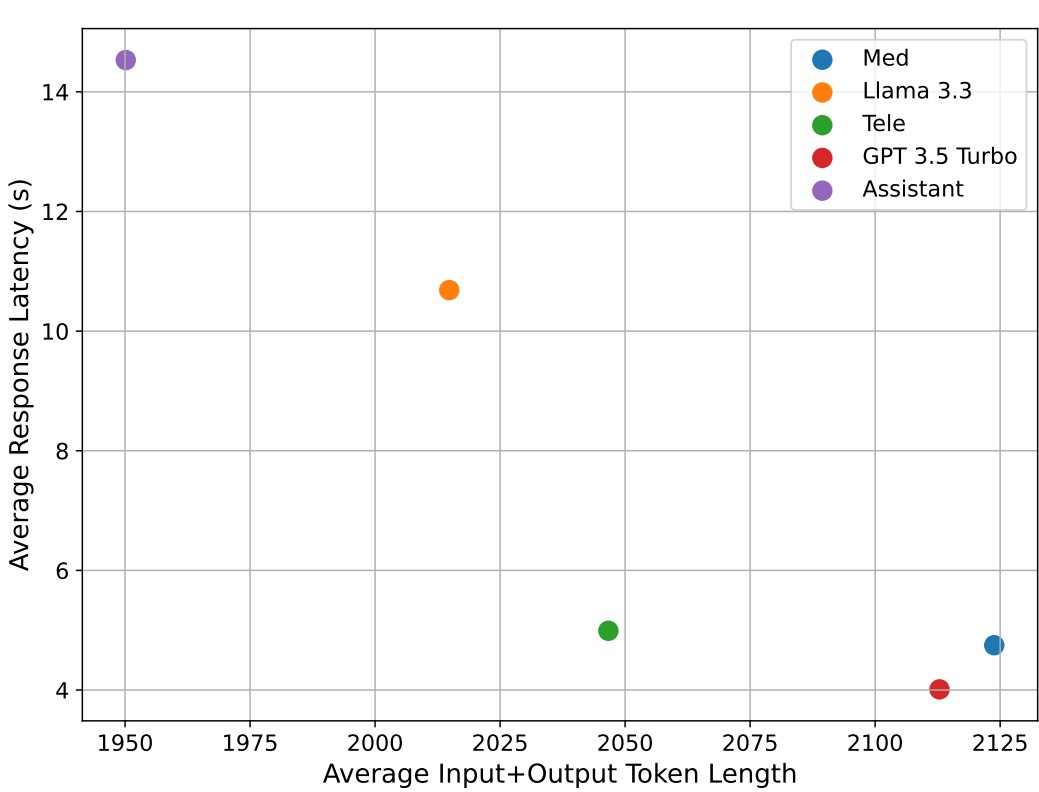

Figure 32: Scatter plot showing average response latency against the sum of the number of input and output tokens for the summarization subtask

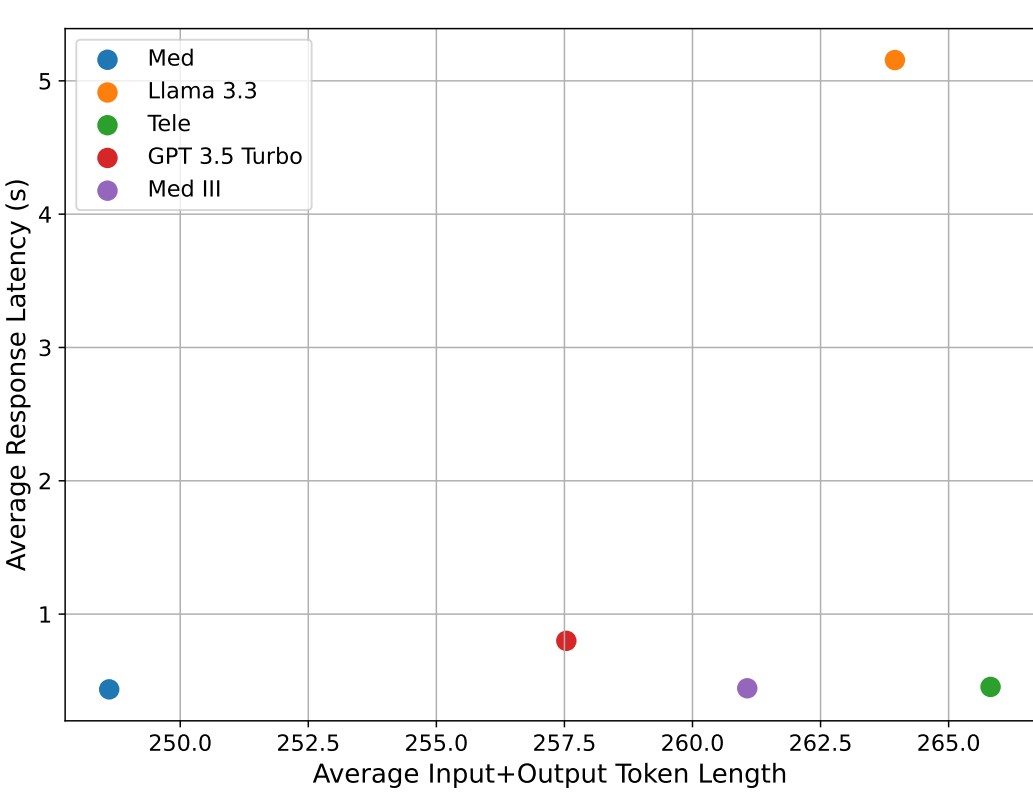

Figure 33: Scatter plot showing average response latency against the sum of the number of input and output tokens for the diagnosis subtask

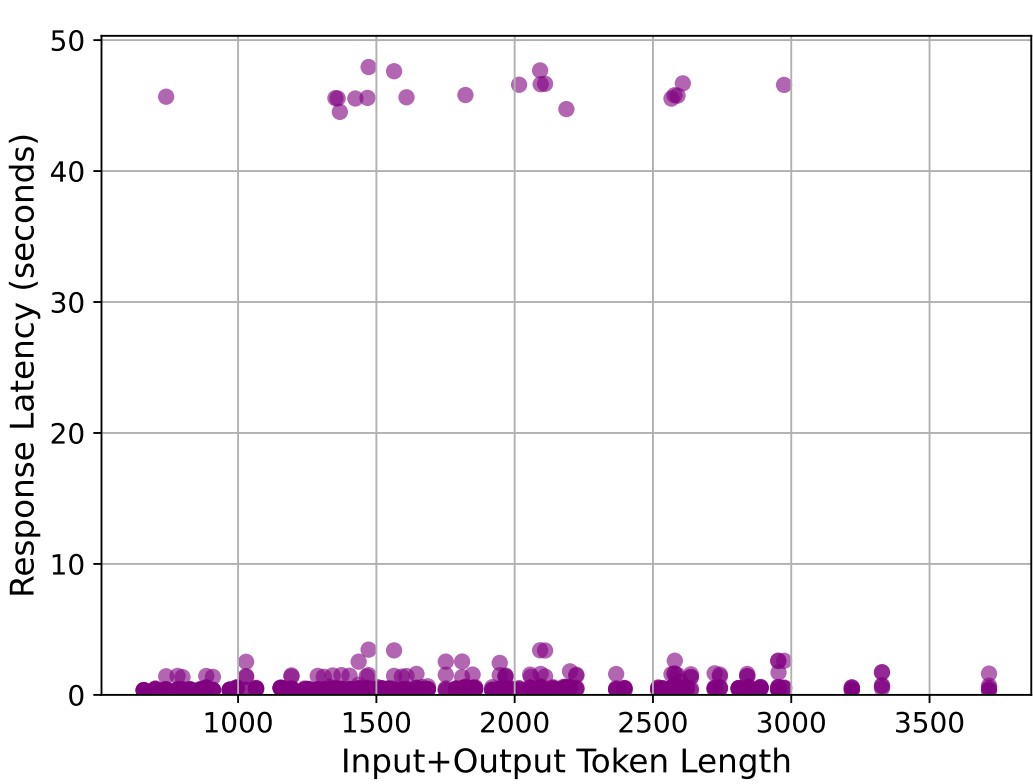

Figure 34: Scatter plot showing average response latency against the sum of the number of input and output tokens for the diagnosis subtask for the Med III model

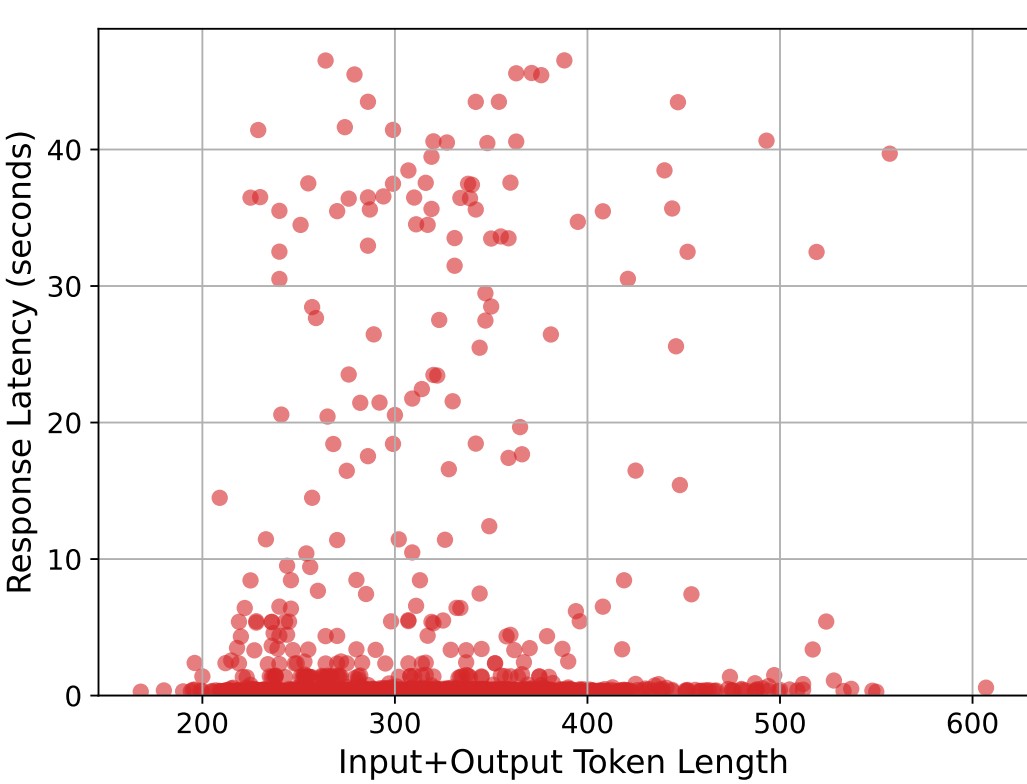

Figure 35: Scatter plot showing average response latency against the sum of the number of input and output tokens for the diagnosis subtask for the GPT 3.5 Turbo model

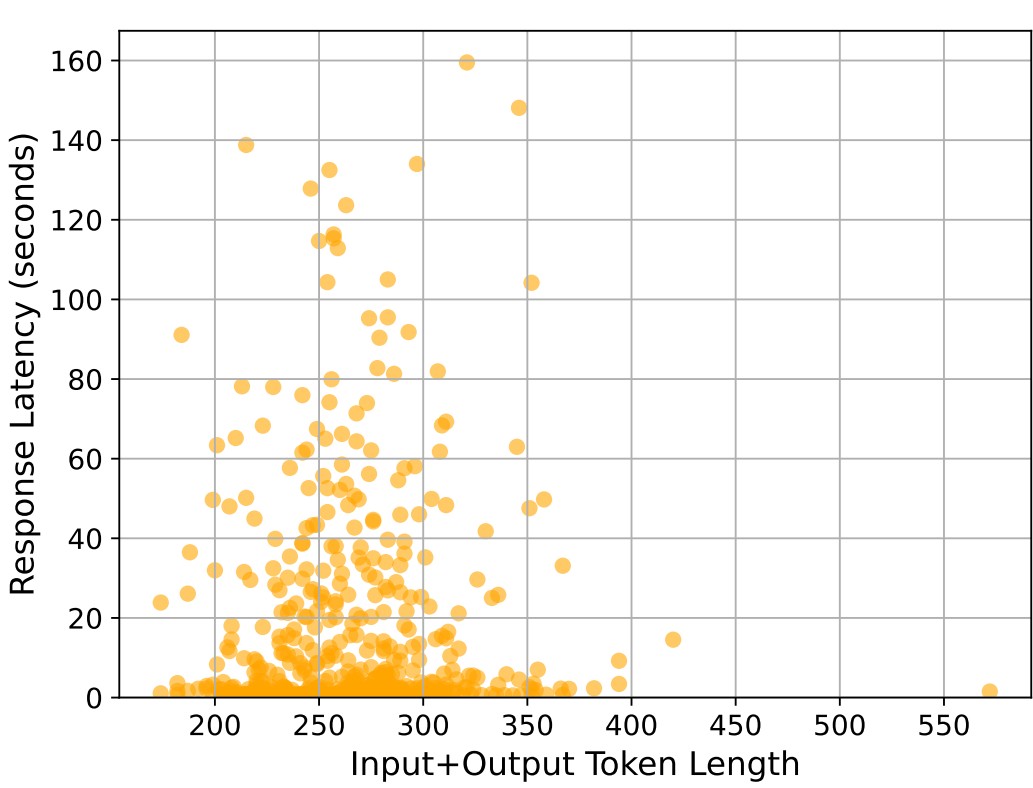

Figure 36: Scatter plot showing average response latency against the sum of the number of input and output tokens for the diagnosis subtask for the Llama model

