# OpenReview forum: "Neural Bandit Based Optimal LLM Selection for a Pipeline of Tasks"
_ICLR.cc/2026/Conference — Submitted to ICLR 2026_

### Official Review · Reviewer_y6C9 · 2025-10-20

**Soundness:** 2
**Presentation:** 3
**Contribution:** 2
**Rating:** 4
**Confidence:** 4

**Summary:**

With the increasing use of large language models (LLMs) for a wide range of tasks, there is growing interest in strategies that can efficiently predict which LLMs will succeed in completing a task, especially in a cost-effective manner. This is becoming even more relevant as platforms like Microsoft and OpenAI offer users the ability to create custom LLM assistants, specialized for specific types of queries. However, some tasks might benefit from breaking them down into smaller subtasks, each of which can be performed by an LLM tailored to handle that specific subtask.

For example, in medical diagnosis extraction, the process could involve selecting one LLM to summarize the medical record, another to validate the summary, and a third to extract the diagnosis from the validated summary. Unlike traditional LLM selection algorithms, this process requires selecting a sequence of LLMs, where each LLM’s output feeds into the next, creating a chain of dependencies that influence the cost and success rate of the task.

The challenge lies in selecting the right sequence of LLMs, where the quality of each subtask’s output directly affects the next LLM's performance. To solve this, the authors propose a neural contextual bandit-based algorithm that learns to model LLM success on each subtask in an online manner, guiding the selection of LLMs without the need for historical performance data. Experiments on datasets related to telecommunications question answering and medical diagnosis prediction show that the proposed algorithm outperforms existing LLM selection algorithms in terms of efficiency and success.

**Strengths:**

1. The use of neural contextual bandits is interesting and this allows the model to adapt to new queries and subtasks dynamically, ensuring better decision-making in LLM selection.

2. The paper introduces the idea of breaking down complex tasks into smaller subtasks and selecting specialized LLMs for each one. This approach leverages the strengths of different LLMs for specific tasks, leading to more efficient and effective solutions.

3. The paper is well-written and clear to read. Abundant theoretical proofs and visualizations are provided to enhance the readability.

**Weaknesses:**

1. it may introduce challenges in terms of training the model efficiently, especially when there is no historical data available. This could make the training process more computationally intensive, particularly for large-scale systems.

2. The idea in the paper is not very novel, as a similar concept is presented in the paper "Cost-efficient knowledge-based question answering with large language models," which also addresses the selection of models in a cost-efficient manner.

3. The need to select a sequence of LLMs for each subtask introduces a layer of decision-making that might introduce delays, especially in time-sensitive applications.

4. It would be helpful to see more generalizable results across a wider range of tasks to assess the broader applicability of the method.

**Questions:**

The model could benefit from more efficient training methods or pre-training strategies that reduce the computational cost of learning to select LLMs for new subtasks.

---

> ### Author Response · Authors · 2025-11-21
>
> We thank the reviewer for their valuable feedback. Below we respond to all of the reviewer's concerns/questions in detail.
> - **It may introduce challenges in terms of training the model efficiently, especially when there is no historical data available.** : In terms of the concern regarding training the model efficiently when there is no historical data available, we want to note that our approach is an online learning approach, which operates in the setting where we have no historical data available and learns to make better decisions over time. Thus, no historical data is needed (indeed, this is one of the advantages of our online approach).
> - **The idea in the paper is not very novel, as a similar concept is presented in the paper “Cost-efficient knowledge-based question answering with LLMs”, which also addresses the selection of models in a cost efficient manner.** : The cited work “Cost-efficient knowledge-based QA with LLMs” (Coke) considers a different problem and formulation from ours. Coke addresses single-step KBQA and selects one model (either a KGM or an LLM) per question, using a cluster-level Thompson sampler and a linear contextual bandit with a cost-regret penalty. In contrast, our paper formulates sequential selection of a pipeline of LLMs for decomposed tasks, where the output of an earlier LLM becomes the input to later subtasks, and both accuracy and monetary cost are defined at the pipeline level. Our Sequential Bandits algorithm is a neural contextual combinatorial bandit with separate neural networks for each (subtask, LLM) pair and an explicit cost term in the UCB objective, designed to learn cross-subtask dependencies in both performance and cost. Thus, even though both works use bandits to trade off accuracy and cost, Coke is a router over KB and LLM models for single-step QA, whereas our contribution is the first to study cost-aware LLM pipeline selection with sequential dependencies between subtasks. We have cited this work in our revised pdf document in lines (136-138).
> - **The need to select a sequence of LLMs for each subtask introduces a layer of decision-making that might introduce delay, especially in time-sensitive applications** : As the reviewer states, the need to select a sequence of LLMs for each subtask can introduce delay. However, we would like to note that the neural networks we use are quite small and shallow with a hidden dimension of 50 and a total parameter count of approximately 19,300 for each of the neural networks. Training these neural networks during model selection takes negligible time compared to the LLM API calls that we make in every iteration. In fact, we conducted some recent experiments on the Telecom dataset where we measured the runtime of a fixed baseline like Random and compared it to the runtime of our algorithm Sequential Bandits. Random achieved an average of 5657 seconds per run averaged over 5 runs for 150 rounds while Sequential Bandits achieved a runtime of 5666 seconds per run averaged over 5 runs for 150 rounds.  We included a paragraph about runtime, these results and a discussion in lines 506-515. Another concern could be the pipeline of LLMs that we are introducing could increase the delay as we are making more API calls compared to just selecting a single LLM for the task at hand, without decomposing it into subtasks. We illustrate the benefit of the pipelined approach in Figure 1, where for the medical setting, including a summarization step in the pipeline before giving a diagnosis improves the diagnosis accuracy for most of the models with the exception of Med III (due to the fact that this model was finetuned on the dataset using the full medical report, not the summary). There is essentially a tradeoff between in terms of better performance and latency when comparing pipelined vs no pipeline settings.
> - **It would be helpful to see more generalizable results across a wider range of tasks to assess the broader applicability of the method.** : We appreciate this suggestion. While additional datasets would strengthen our evaluation, we believe our current experiments demonstrate the method's effectiveness across meaningfully diverse settings. Specifically, we evaluate on two domains requiring specialized knowledge (medical and telecommunications) with fundamentally different task structures: one involving multiple-choice question answering and the other requiring open-ended generation. Our choice of datasets is constrained by the method's requirements, sequential task structure and subtask dependencies, which many standard benchmarks lack. More importantly, we contend that the primary benefit of our approach lies in complex, domain-specific tasks where decomposition into dependent subtasks is beneficial. The datasets we selected are representative of this target use case.

---

### Official Review · Reviewer_CrjG · 2025-11-01

**Soundness:** 3
**Presentation:** 3
**Contribution:** 2
**Rating:** 4
**Confidence:** 4

**Summary:**

This paper addresses **cost-effective LLM selection for tasks decomposed into sequential pipelines**, where different LLMs handle each subtask and outputs feed forward. This creates complex interdependencies: earlier LLM choices affect input quality and optimal selections for downstream subtasks.

The authors propose **Sequential Bandits**, a neural contextual bandit algorithm with two key innovations: (1) maintaining **separate neural networks for each (subtask, LLM) pair** to model rewards, and (2) **sequentially** selecting LLMs after observing previous outputs rather than committing to all choices simultaneously. The UCB-based selection strategy balances exploration-exploitation while incorporating costs via tunable parameters $\alpha_i$.

**Main contributions** include: (1) novel problem formulation for sequential LLM pipeline selection, (2) the Sequential Bandits algorithm grounded in neural contextual bandits, (3) a new medical diagnosis dataset from MIMIC-III, and (4) experiments showing 7.6% and 6.5% improvements over baselines on medical and telecommunications tasks. The work assumes task decomposition is pre-defined rather than learned.

**Strengths:**

**Originality:** The paper identifies a new problem that sits between existing LLM routing and cascading work. The key insight—that LLM choices for earlier subtasks affect optimal selections downstream—is well-motivated.

**Quality:** The experimental methodology demonstrates careful design in several respects. The baselines are thoughtfully constructed to isolate specific contributions: Random shows learning value and fixed Llama/Tele shows adaptation value.

**Clarity:** The paper is well-written with clear motivation and straightforward presentation. Figure 2 effectively illustrates the problem, and Algorithm 1 is easy to follow.

**Significance:** The problem is timely given the rise of agentic AI and custom LLM assistants. The framework is general and shows meaningful practical improvements.

**Weaknesses:**

**Limited Baseline Comparisons:** The most significant experimental weakness is the absence of comparisons to recent LLM routing and cascading methods extensively cited in related work, such as FrugalGPT, Hybrid LLM, and many other routing/cascading frameworks. While many routing methods require historical training data—and learning without such data is indeed a key motivation for the online approach—several evaluation strategies could address this gap: (1) provide a warm-start period where methods collect initial training data before comparison, or (2) evaluate after the online learning phase to compare converged performance. More fundamentally, demonstrating when and by how much online learning outperforms offline-trained routers (in terms of sample efficiency, final performance, or adaptability) would significantly strengthen the contribution. Without these comparisons, readers cannot assess whether the observed improvements come from genuinely superior algorithmic design.

**Task Decomposition Assumption:** The paper assumes task decomposition is given, which is a major practical limitation explicitly acknowledged but not addressed. In real deployments, determining *how* to break down tasks is often harder than selecting models. The medical example (summarize → diagnose) seems intuitive, but for arbitrary complex queries, the optimal decomposition is non-obvious and likely query-dependent. The paper provides no guidance on when pipelines help versus hurt, or how to design good decompositions.

**Lack of Theoretical Analysis:** While the paper adapts NeuralUCB's framework, it provides no formal regret bounds for the sequential setting. The theoretical contribution is essentially "use existing neural bandit algorithms but with separate networks per arm." It's unclear whether standard NeuralUCB regret guarantees transfer to this setting where: (1) contexts change dynamically based on previous LLM outputs, (2) subtask rewards are interdependent, and (3) the super-arm reward is a complex function of base rewards. The exploration-exploitation trade-off in sequential decisions differs fundamentally from standard bandits.

**Limited Experimental Scale and Scope:** The evaluation uses only 100 medical reports and 100 rounds of online learning, which is quite modest. More critically, there's no analysis of: (1) how performance scales with pipeline depth (only 2-3 subtasks tested), (2) sensitivity to the cost parameter α (no ablation shown), (3) what happens with more LLMs per subtask (experiments use 5-7 total models), or (4) performance in non-stationary settings where LLM capabilities or costs change. The token length prediction model's accuracy isn't reported—if predictions are poor, the cost-aware objective could be misleading.

**Cost Model Limitations:** The cost model assumes costs are proportional to tokens, but ignores latency, which can be a critical "cost" in interactive applications. The token predictor is trained on LMSYS-Chat-1M, which may not generalize well to specialized domains (medical, telecom). The fixed token predictor never adapts during online learning, potentially leading to systematic errors.

**Questions:**

See weakness above

---

> ### Author Response · Authors · 2025-11-21
>
> We thank the reviewer for their valuable feedback. Below we respond to all of the reviewer's concerns/questions in detail.
> - **Limited Baseline Comparisons** : We have compared our approach to another LLM selection baseline, FrugalGPT (which uses a model cascade) for the Medical experiments and included the results in Figure 3. FrugalGPT is an offline cascade-building method, and we re-implemented it within our own prompting setup, Azure API calls, evaluation pipeline, and datasets so that comparisons would be fair. Specifically, we replaced FrugalGPT’s built-in completion and scoring components with our Azure inference wrapper and our task-specific reward function. We also added a round-based evaluation loop that applies the fixed cascade to each query in sequence, which lets us compute cumulative reward and cost curves, even though FrugalGPT itself does not perform online learning. Finally, because different cascades depend on the allowed budget, we standardized this by limiting the budget to three times the diagnosis subtask cost for all FrugalGPT variants. We also aim to add the results for the Telecom experiments before the end of the discussion period. These results are for the non-pipelined version of FrugalGPT (which is why FrugalGPT obtains the lowest cost among all baselines in Figure 3), and we will also try to extend it to include a pipelined version before the end of the discussion period and share the results once they are available.
> - **Task Decomposition Assumption** : We fully acknowledge that we do not address how the task decomposition should be done in this paper. However, our goal in this work was to study the decision problem that arises once a task decomposition is specified to us and not to solve the task decomposition itself. Automatic decomposition is indeed an important problem, but we believe that is a separate research direction. In fact, there are some works which focus on solving task decomposition for LLMs such as “ADaPT: As-Needed Decomposition and Planning with Language Models;” these algorithms could be applied to a given task before our sequential bandits approach is used to select which LLMs to use for each subtask in the decomposition. We cited this paper in the related work section (172-175). We do believe that an interesting direction of future work includes incorporating a framework similar to ADaPT in the online training loop of our sequential bandits algorithm, and we briefly discuss this idea in our paper’s conclusion. Regarding the concern that using a pipeline may hurt task performance compared to not breaking a task into subtasks, we agree that this is a possibility. However, it is unlikely to be the case for complex tasks; indeed, the potential benefit of this approach motivates much of the existing literature on task decomposition for LLMs. Since our approach is an online learning approach, our algorithm is able to learn whether a step in the pipeline provides a benefit in terms of task success. If it does not, based on our selection rule, the cost term will govern the model selections, prioritizing cheaper models for that step as the output of this subtask does not significantly contribute to task success.
> - **Lack of Theoretical Analysis** : As mentioned by the reviewer, we do not have any theoretical guarantees in this work, and we agree that this is an important direction of future work. However, we believe this is a nontrivial future direction meriting its own paper, as the typical NeuralUCB guarantees do not easily translate to our sequential pipeline setting. One of the key challenges is that the context of the next arm to be selected in the pipeline depends on the arm choice in the previous step of the pipeline, which effectively makes part of the decision context (i.e., the context of the next arm to be selected) dependent on previous decisions. In standard contextual bandit and neural bandit settings, this is not the case, and we have not found a straightforward way to incorporate such dependence. Our problem does have analogues in the combinatorial bandit setting, in the sense that selecting multiple arms (models) for the different subtasks at the beginning of the task can be modeled as a combinatorial bandit problem, but in this formulation an arm choice made in that round does not affect another arm’s choice. In addition, the selection rule of our algorithm is different from other neural bandit works, as it also includes a cost term which is also updated over time, presenting another challenge.

---

> > ### Author Response · Authors · 2025-11-21
> >
> > - **Limited Experimental Scale and Scope** : We also acknowledge that 100 rounds of online learning is quite modest and have increased the number of rounds we run the Telecom experiments to 150 rounds instead. We do not find any qualitative differences in the resulting performance, as shown in Figure 3. Before the end of the rebuttal period, we aim to also increase the number of rounds of the Medical experiments from 100 to 150, and we will share those results when they are available. We have some results on how performance scales with pipeline depth as we experimented with several different pipelines. We have tested 2 and 3 subtask pipelines for the medical setting as well as single task and 3 subtask pipelines for the telecom experimental setting. We include the results for these settings in the Appendices A and C. We acknowledge that experiments on longer pipelines would help to further validate our approach. However, since this is to the best of our knowledge the first work to investigate LLM selection for a pipeline of dependent subtasks, there are no standard datasets for longer pipelines; indeed, we had to modify the existing medical dataset to run our experiments even with 3 task pipelines, as described in Section 5.2 and Appendix section B.2. Thus, we expect that validating our work on longer pipelines would require significant dataset engineering. Regarding the sensitivity to the cost parameter $\alpha$, we acknowledge that we do not have an ablation for this parameter, however, we have some results on cost agnostic settings (when $\alpha=0$) in the Appendix section A. We expect our algorithm to scale well even if there are more LLMs per subtask due to the fact that the compute of our algorithm is independent of the number of LLMs per subtask, as a single LLM is chosen for each subtask in any given round and only that chosen LLM’s neural network is trained. We expect our algorithm to perform well even for a larger number of LLMs for each subtask, given that we also run the algorithm for a larger number of rounds. A constraint which prevented us from adding more LLMs for the subtasks was that the LMSYS-Chat-1M dataset does not include some of the commonly used LLMs, and also it is not possible to make API calls for some of the models in this dataset. We further believe that our algorithm will be able to adapt to nonstationary changes over time as it is an online learning approach and can utilize the feedback it receives from LLMs to initiate changes to its decision making. We have included some figures in the Appendix showing the error of the token length prediction model after being trained offline and offline+online (Figures 10-13).
> > - **Cost Model Limitations** : We fully agree that latency is an important objective to consider as a cost, especially in interactive applications, and we have included some results for some experiments we conducted about the relation of input and output prompt length to response latency for different LLM models in Appendix D. The challenge in incorporating latency as a cost in our objective is that to our knowledge, there are no works which propose a predictive model that gives latency estimations for models based on the input prompt length, which is what we would need to incorporate latency into our objective. Our experimental results in Appendix D for different models have also shown us that response latency is highly variable and often hard to predict. The API calls made for commonly used models like Llama and GPT 4 show more variation than the finetuned models, which is likely due to the variable traffic in these models.
> > In experiments we conducted recently, we have also trained our cost predictor model in an online manner using the number of tokens in the LLM outputs so that if an LLM changes to become more or less verbose, our cost predictor can adjust as well. Regarding the fixed cost model during training, we have changed it to become an online model rather than a fixed one by training it using the feedback we get (lines 10 and 16 in the updated Algorithm 1). We include some results and explanations of these results for the online cost model in the updated pdf document (Appendix section B.3). As the reviewer mentioned, results on the offline trained cost model showed larger errors for specialized domains like medical with Med III having higher error (Figure 11), however after training it online, the errors have decreased significantly for the Med III and Tele models (Figure 12).

---

### Official Review · Reviewer_23oe · 2025-11-01

**Soundness:** 2
**Presentation:** 2
**Contribution:** 2
**Rating:** 2
**Confidence:** 2

**Summary:**

This paper proposed Sequential Bandits, a contextual MAB algorithm variant adapted to selecting a pipeline of LLMs for each subtasks in task decomposing paradigm. Specifically, Sequential Bandits is designed for the scenario of complicated tasks that cannot be well-solved in single step by one LLM alone and rely on selecting suitable LLMs to perform decomposed subtasks in order to achieve the optimal results. Different from traditional neural MAB methods, Sequential Bandits employs separate reward neural networks for each LLM combinations to enrich the exploration while keeping the training overhead. Experiments on MIMIC-III and TeleQnA datasets exhibit Sequential Bandits has a better net reward and a lower economic cost compared to other adapted traditional MAB methods and random baselines.

**Strengths:**

S1: This paper first formulate selecting LLM pipeline into an instance of the classical contextual MAB problem.

S2: The method design and experiment considered cost-effectiveness during LLM selection pipeline.

S3: Comparison with naively adapted traditional MAB baselines is conducted to show the effectiveness of separate neural network design in Sequential Bandits.

**Weaknesses:**

I'm not expert in online LLM selection technique and MAB algorithm, thus I can only provide some literal and experiment suggestions for the paper. I hope the other knowledgeable reviewers can provide more technical feedback.

W1: Fig 2 could provide more elaborated or distinguished details for Sequential Bandits to make it friendly to the audiences from other fields.

W2: As provided in related works, it seems there exist other recent studies for LLM selection pipeline, while the experiment only employed adapted traditional MAB baselines.

W3: Although illustrate reward and cost for each baseline, a more detailed results on task accuracy, success rate, etc. are not provided to give a more clear review of the method effectiveness or usability.

**Questions:**

See Weaknesses.

---

> ### Author Response · Authors · 2025-11-21
>
> We thank the reviewer for their valuable feedback. Below we respond to all of the reviewer's concerns/questions in detail.
> - **Fig 2 could provide more elaborated or distinguished details for Sequential Bandits to make it friendly to the audiences from other fields.** : We have updated Figure 2 to make it more friendly for readers from other fields by simplifying the language and also adding the online cost model training. We would be happy to make further revisions if the reviewer thinks that any aspects of the revised figure are confusing or difficult to understand for non-experts.
> - **As provided in related works, it seems there exist other recent studies for LLM selection pipeline, while the experiment only employed adapted traditional MAB baselines.** : We have compared our approach to another LLM selection baseline, FrugalGPT (which uses a model cascade) for the Medical experiments and included the results in Figure 3. FrugalGPT is an offline cascade-building method, and we re-implemented it within our own prompting setup, Azure API calls, evaluation pipeline, and datasets so that comparisons would be fair. Specifically, we replaced FrugalGPT’s built-in completion and scoring components with our Azure inference wrapper and our task-specific reward function. We also added a round-based evaluation loop that applies the fixed cascade to each query in sequence, which lets us compute cumulative reward and cost curves, even though FrugalGPT itself does not perform online learning. Finally, because different cascades depend on the allowed budget, we standardized this by limiting the budget to three times the diagnosis subtask cost for all FrugalGPT variants. We also aim to add the results for the Telecom experiments before the end of the discussion period. These results are for the non-pipelined version of FrugalGPT (which is why FrugalGPT obtains the lowest cost among all baselines in Figure 3), and we will also try to extend it to include a pipelined version before the end of the discussion period and share the results once they are available.
> - **Although illustrate reward and cost for each baseline, a more detailed results on task accuracy, success rate, etc. are not provided to give a more clear review of the method effectiveness or usability.** : We have included some results about task accuracy in the Appendix other than Figure 1 in the main paper, which we detail below. The task accuracy for the medical dataset is defined according to whether the LLM pipeline successfully identified a correct diagnosis, and that for the telecom data is defined according to whether the LLM pipeline successfully identified the correct answer to a multiple-choice question and explained the answer it gave correctly. Figure 17 shows the average accuracy of models for the diagnosis subtask in 3 subtask medical setting given the summarizer is selected at random, Figure 19-21 display the average accuracy of models for the summarization, answering and explanation subtasks given all models are selected at random (equivalent to Random baseline) for the Telecom setting. Figure 25 shows the average accuracy of models for the summarization subtask given the summarizer is selected at random for the medical setting. Figure 26 shows the diagnosis accuracies given the summarizer is selected as the model shown for random diagnoser selection for the medical setting and Figure 27 is a plot of the average accuracy of models for the diagnosis subtask for random summarizer and diagnoser selection in the medical setting.

---

### Official Review · Reviewer_4YAL · 2025-11-01

**Soundness:** 4
**Presentation:** 4
**Contribution:** 4
**Rating:** 6
**Confidence:** 2

**Summary:**

The paper proposes a novel approach called "Sequential Bandits," which adapts the contextual multi-armed bandit (MAB) algorithm to select the best large language models (LLMs) for sequential, multi-step tasks. The main idea is to model the task decomposition as a pipeline of subtasks, each tackled by a different LLM. The algorithm aims to maximize the accuracy of task completion while also minimizing costs, providing a framework for intelligent LLM selection in complex tasks that cannot be handled by a single model. The proposed method is evaluated on medical diagnosis prediction and telecommunications question answering tasks.

**Strengths:**

The paper introduces the "Sequential Bandits" algorithm, which effectively addresses the problem of selecting the best LLMs for multi-step tasks, providing a solution that is more practical than existing methods.Experiments show that the proposed method outperforms baseline algorithms in medical diagnosis and telecommunications question answering tasks, demonstrating its ability to balance accuracy and cost.

**Weaknesses:**

The details of the cost prediction model, particularly in terms of output token length prediction and overall cost estimation, are not fully explained. The paper compares the method to a few baselines but does not offer a comprehensive comparison with other state-of-the-art LLM selection algorithms.The approach uses separate neural networks for each subtask, which could lead to scalability challenges for larger systems.

**Questions:**

The details of the cost prediction model, particularly in terms of output token length prediction and overall cost estimation, are not fully explained. The paper compares the method to a few baselines but does not offer a comprehensive comparison with other state-of-the-art LLM selection algorithms.The approach uses separate neural networks for each subtask, which could lead to scalability challenges for larger systems.

---

> ### Author Response · Authors · 2025-11-21
>
> We thank the reviewer for their valuable feedback. Below we respond to all of the reviewer's concerns/questions in detail.
> - **The details of the cost prediction model, particularly in terms of output token length prediction and overall cost estimation, are not fully explained.** : We would like to note that we give details about the cost prediction model, output token length prediction and overall cost estimation in lines 249-254, 387-394 in the main paper and sections B.3 and B.4 in the Appendix (lines 1115-1151, 1178-1187). We have moved some of these details to the main body of the paper (lines 387-394) and are happy to provide further details if desired.
> - **The paper compares the method to a few baselines but does not offer a comprehensive comparison with other state-of-the-art LLM selection algorithms.** : We have compared our approach to another LLM selection baseline, FrugalGPT (which uses a model cascade) for the Medical experiments and included the results in Figure 3. FrugalGPT is an offline cascade-building method, and we re-implemented it within our own prompting setup, Azure API calls, evaluation pipeline, and datasets so that comparisons would be fair. Specifically, we replaced FrugalGPT’s built-in completion and scoring components with our Azure inference wrapper and our task-specific reward function. We also added a round-based evaluation loop that applies the fixed cascade to each query in sequence, which lets us compute cumulative reward and cost curves, even though FrugalGPT itself does not perform online learning. Finally, because different cascades depend on the allowed budget, we standardized this by limiting the budget to three times the diagnosis subtask cost for all FrugalGPT variants. We also aim to add the results for the Telecom experiments before the end of the discussion period. These results are for the non-pipelined version of FrugalGPT (which is why FrugalGPT obtains the lowest cost among all baselines in Figure 3), and we will also try to extend it to include a pipelined version before the end of the discussion period and share the results once they are available.
> - **The approach uses separate neural networks for each subtask, which could lead to scalability challenges for larger systems.** : We agree that the scalability challenges for larger systems is a valid concern and important point. In particular, scaling to tasks with a large number of subtasks may appear to be a significant challenge, as we train separate neural networks for each subtask. However, we would like to note that the neural networks we use are quite small and shallow with a hidden dimension of 50 and a total parameter count of approximately 19,300 for each of the neural networks. Thus, keeping track of the network parameters for each subtask is likely to have tolerable memory and storage overhead. Training these neural networks during model selection also takes negligible time compared to the LLM API calls that we make in every iteration. In fact, we conducted some recent experiments on the Telecom dataset where we measured the runtime of a fixed baseline like Random (which does not run any sophisticated LLM selection logic) and compared it to the runtime of our algorithm Sequential Bandits. Random achieved an average of 5657 seconds per run averaged over 5 runs while Sequential Bandits achieved a comparable runtime of 5666 seconds per run averaged over 5 runs.  We include a paragraph about runtime and scalability in the updated paper, including these results and a discussion (lines 506-515).

---

### Author Response · Authors · 2025-12-03

We have extended our experiments to include the FrugalGPT baseline and ran the experiments for 150 rounds instead of 100 rounds for both telecom and medical settings. These results could be found in the updated pdf document.

---

### Meta-Review · Area_Chair_kve7 · 2026-01-04

**Summary:**

The core concerns of the reviewers focus on four main areas: insufficient baseline comparisons (lack of comparison with SOTA LLM selection methods), lack of theoretical analysis, limitations of the task decomposition assumption, and limited experimental scale and scalability.

**Reviewer Concerns:**

The authors provided effective responses regarding baseline comparisons (adding FrugalGPT experiments) and details of the cost prediction model; however, issues such as the absence of theoretical analysis, inherent limitations of the task decomposition assumption, and insufficient experimental depth and generalizability remain unresolved.

**Reviewer Scores:**

If they had participated fully in the discussion, Reviewer 4YAL (score 6) might have raised the score due to the added FrugalGPT comparison; Reviewer CrjG (score 4) might have maintained a score of 4 or slightly lowered it to 2 due to the theoretical shortcomings and experimental limitations; Reviewer y6C9 (score 4) might have kept the score at 4, as concerns about innovation remain.

---

### Decision · Program_Chairs · 2026-01-26

Reject